EMBO
Molecular Medicine

# A KDM6A–KLF10 reinforcing feedback mechanism aggravates diabetic podocyte dysfunction

Chun-Liang Lin[1,2,3,4,5], Yung-Chien Hsu[1,4], Yu-Ting Huang[1,4], Ya-Hsueh Shih[1,4], Ching-Jen Wang[5,6], Wen-Chih Chiang[7] & Pey-Jium Chang[1,4,8,*] (ID)

## Abstract

Diabetic nephropathy is the leading cause of end-stage renal disease. Although dysfunction of podocytes, also termed glomerular visceral epithelial cells, is critically associated with diabetic nephropathy, the mechanism underlying podocyte dysfunction still remains obscure. Here, we identify that KDM6A, a histone lysine demethylase, reinforces diabetic podocyte dysfunction by creating a positive feedback loop through up-regulation of its downstream target KLF10. Overexpression of KLF10 in podocytes not only represses multiple podocyte-specific markers including nephrin, but also conversely increases KDM6A expression. We further show that KLF10 inhibits nephrin expression by directly binding to the gene promoter together with the recruitment of methyltransferase Dnmt1. Importantly, inactivation or knockout of either KDM6A or KLF10 in mice significantly suppresses diabetes-induced proteinuria and kidney injury. Consistent with the notion, we also show that levels of both *KDM6A* and *KLF10* proteins or mRNAs are substantially elevated in kidney tissues or in urinary exosomes of human diabetic nephropathy patients as compared with control subjects. Our findings therefore suggest that targeting the KDM6A–KLF10 feedback loop may be beneficial to attenuate diabetes-induced kidney injury.

**Keywords** diabetic nephropathy; epigenetics; KDM6A; KLF10; podocyte dysfunction

**Subject Categories** Metabolism; Urogenital System

## Introduction

Diabetic nephropathy is the most common cause of end-stage renal disease (Collins *et al*, 2014). Although a growing body of evidence shows that injury of podocytes, also known as glomerular visceral epithelial cells, critically contributes to the progression of diabetic nephropathy (Wiggins, 2007; May *et al*, 2014), the molecular determinants underlying podocyte dysfunction are still poorly understood. Besides genetic mutations, aberrant epigenetic alterations such as DNA methylation or histone modification may play a pivotal role in causing podocyte injury (Feliers, 2015; Majumder & Advani, 2015). Previous studies have shown that treatment with 5-aza-2′-deoxycytidine, a DNA demethylating agent, substantially increased levels of podocyte-specific markers such as nephrin and Neph3 in human immortalized podocytes or human embryotic kidney cell lines (Ristola *et al*, 2012). On the other hand, positive or negative actions of various histone deacetylases (HDACs) such as SIRT1, SIRT6, HDAC2, HDAC4, and HDAC9 as well as histone acetyltransferases (HATs) such as GCN5 have been linked to podocyte dysfunction in diabetic nephropathy (Noh *et al*, 2009; Bock *et al*, 2013; Wang *et al*, 2014; Yacoub *et al*, 2014; Liu *et al*, 2016, 2017). Additionally, histone methylation and demethylation can also influence the expression of a wide variety of protective and pathogenic genes in the development of diabetic nephropathy. Generally, methylation at histone H3 lysine 4 (H3K4), lysine 36 (H3K36), and lysine 79 (H3K79) is associated with gene expression, whereas methylation at histone H3 lysine 9 (H3K9) and lysine 27 (H3K27) as well as methylation at histone H4 lysine 20 (H4K20) is associated with gene repression. Previously, Lefevre *et al* (2010) have shown that loss of an essential component PTIP of the H3K4 methyltransferase complex in podocytes caused a slowly progressing proteinuria. More recently, Majumder *et al* (2018) reported that loss of the histone H3 lysine 27 trimethylation (H3K27me3) mark in podocytes by inactivation or deletion of EZH2, a H3K27 methylating enzyme, also resulted in podocyte dedifferentiation and glomerular injury. Conversely, pharmacological inhibition of two demethylating enzymes KDM6A (also called UTX) and KDM6B (also called JMJD3), which specifically demethylate H3K27 dimethylation (H3K27me2) and H3K27 trimethylation (H3K27me3), attenuated glomerular disease (Majumder *et al*, 2018). Despite the importance of various histone-modifying enzymes

1    Departments of Nephrology, Chang Gung Memorial Hospital, Chiayi, Taiwan
2    Kidney Research Center, Chang Gung Memorial Hospital, Taipei, Taiwan
3    College of Medicine, Chang Gung University, Taoyuan, Taiwan
4    Kidney and Diabetic Complications Research Team (KDCRT), Chang Gung Memorial Hospital, Chiayi, Taiwan
5    Center for Shockwave Medicine and Tissue Engineering, Department of Medical Research, Chang Gung Memorial Hospital, Kaohsiung, Taiwan
6    Department of Orthopedic Surgery, Chang Gung Memorial Hospital, Kaohsiung, Taiwan
7    Department of Internal Medicine, National Taiwan University Hospital, Taipei, Taiwan
8    Graduate Institute of Clinical Medical Sciences, College of Medicine, Chang Gung University, Taoyuan, Taiwan
     *Corresponding author. Tel: +886 5 362 1000; E-mail: peyjiumc@mail.cgu.edu.tw

and DNA methylation patterns in modulating podocyte function, the detailed connection between the global epigenetic reprogramming and the resultant signal transduction or gene expression under diabetic conditions remains obscure.

Kruppel-like factors (KLFs) are evolutionarily conserved and structurally related zinc finger-containing transcription factors, which also share sequence homology with specificity proteins such as SP1 and SP3 (McConnell & Yang, 2010; Bialkowska et al, 2017). Like SP subfamily members, KLFs mediate transcriptional activation and/or repression by recognizing similar GC-rich consensus sequences in target DNA, along with the recruitment of specific co-activators, co-repressors, or other chromatin remodeling proteins (McConnell & Yang, 2010; Bialkowska et al, 2017). Accumulating evidence has implicated that KLFs play a vital role in multiple biological processes and diseases. For instance, several reports have shown that loss of specific KLF members including KLF2, KLF4, KLF6, and KLF15 in glomerular endothelial cells or podocytes resulted in exacerbation of kidney injury under various stress conditions (Mallipattu et al, 2012, 2015, 2017; Hayashi et al, 2014, 2015; Zhong et al, 2015, 2016). These results indicate that all these reported KLF members act to protect kidney from damage. To date, there is no evidence showing whether other KLF family members in these cell types have opposite actions and contribute to kidney dysfunction.

In this study, we report a new deleterious pathway driven by a positive feedback loop involving KDM6A and KLF10, a SP/KLF member originally identified as a TGF-β-inducible early gene product (Subramaniam et al, 1995), in podocytes under diabetic conditions, and suggest that inhibition of the regulatory pathway may help prevent progression of diabetic nephropathy.

# Results

### Linking epigenetic modifications to podocyte dysfunction

In order to explore the functional roles of epigenetic regulation such as DNA methylation or histone modification in podocyte dysfunction, we initially used nephrin as a marker to study its regulated expression in immortalized mouse podocytes. Treatment of cultured podocytes with high glucose (HG), but not high mannitol (Mann), decreased nephrin expression (Fig 1A), which was correlated with increased DNA methylation at the nephrin gene promoter as determined by methylation-specific PCR (Fig 1B and Appendix Fig S1). On the other hand, treatment with pargyline hydrochloride, a pan-KDM inhibitor, prevented the effects of HG on nephrin expression and its promoter DNA methylation (Fig 1C and D), suggesting that certain KDM enzymes directly or indirectly modulate nephrin expression. After examining mRNA levels of various KDMs in podocytes, our results showed that HG treatment significantly up-regulated KDM6A, but not KDM3A, KDM4A, KDM5A, KDM5B, and KDM6B (Fig 1E). KDM6A, also named UTX, is an X-linked protein that functions to demethylate H3K27me3 and H3K27me2, two repressive histone markers (Wang et al, 2010; Rotili & Mai, 2011). Immunoblotting analysis demonstrated that increased KDM6A expression, along with reduced levels of H3K27me3 and H3K27me2, was detected specifically in HG-treated podocytes, but not in mannitol-treated podocytes (Fig 1F and G). Knockdown and overexpression of KDM6A in podocytes further supported the notion that increased KDM6A expression critically contributed to podocyte dysfunction, showing decreased levels of nephrin, WT-1 (another podocyte-specific protein), and H3K37me3, as well as increased levels of DNA methylation at the nephrin promoter (Fig 2A–D). Consistently, treatment with a KDM6A inhibitor, GSK-J4, halted HG-induced reduction of nephrin, WT-1 and H3K27me3 expression (Fig 2E) and HG-induced DNA methylation at the nephrin promoter (Fig 2F). Collectively, our results highlighted that increased KDM6A expression endangers podocyte function.

### Elucidating the role of KDM6A in diabetes-induced kidney injury

We next investigated the in vivo expression and significance of KDM6A in diabetic kidneys using a streptozotocin (STZ)-induced mouse model. We indeed found that higher levels of KDM6A, accompanied by lower levels of nephrin, WT1 and H3K27me3, were significantly expressed in kidney tissues of the 4-, 8-, and 12-week diabetic mice relative to normal mice (Fig 3A and B). In these STZ-treated mice, several important signs of diabetic nephropathy

**Figure 1. Multi-layered epigenetic controls contribute to nephrin down-regulation in immortalized mouse podocytes.**

A  Western blot analysis of nephrin expression in immortalized podocytes cultured in normal (5 mM) or high glucose (HG; 30 mM) or high mannitol (Mann; 5 mM glucose plus 25 mM mannitol) for 24, 48, and 72 h. *$P < 0.05$ versus normal controls for the indicated time points (Wilcoxon two-sample test; $n = 3$).

B  Analysis of the methylation status at nephrin gene promoter in podocytes cultured in normal, high glucose, or high mannitol at the indicated time points. U, unmethylated-specific primers; M, methylated-specific primers. Experiments were repeated three times, and a representative gel from one experiment is shown.

C  Effect of pargyline hydrochloride, a pan-KDM inhibitor, on high glucose-mediated nephrin expression. Immortalized mouse podocytes were cultured in the indicated cultured media in the presence or absence of pargyline hydrochloride (1 μM) for 48 h. Protein lysates from the treated podocytes were analyzed for nephrin expression by immunoblotting. *$P < 0.05$ versus normal controls, #$P < 0.05$ versus untreated HG-incubated cells (Wilcoxon two-sample test; $n = 3$).

D  Effect of pargyline hydrochloride on high glucose-mediated DNA methylation at nephrin gene promoter. Genomic DNA samples from the treated podocytes in (C) were analyzed for methylation status of nephrin gene promoter by methylation-specific PCR. Experiments were repeated three times, and a representative gel from one experiment is shown.

E  Quantitative RT–PCR analyses of different KDMs in immortalized podocytes exposed to high glucose or high mannitol for 48 h. *$P < 0.05$ versus normal controls (Wilcoxon two-sample test; $n = 3$).

F  Western blot analysis of KDM6A expression in podocytes cultured in normal, high glucose, or high mannitol conditions for 24, 48, and 72 h. *$P < 0.05$ versus normal controls for the indicated time points (Wilcoxon two-sample test; $n = 3$).

G  Western blot analysis of H3K27me1, H3K27me2, H3K27me3, and pan-methyl H3K9 in podocytes cultured in normal, high glucose, or high mannitol conditions for 9 and 24 h. *$P < 0.05$ versus normal controls (Wilcoxon two-sample test; $n = 3$).

Data information: Data are expressed as mean ± SEM. See the exact P-values for comparison tests in Appendix Table S1.

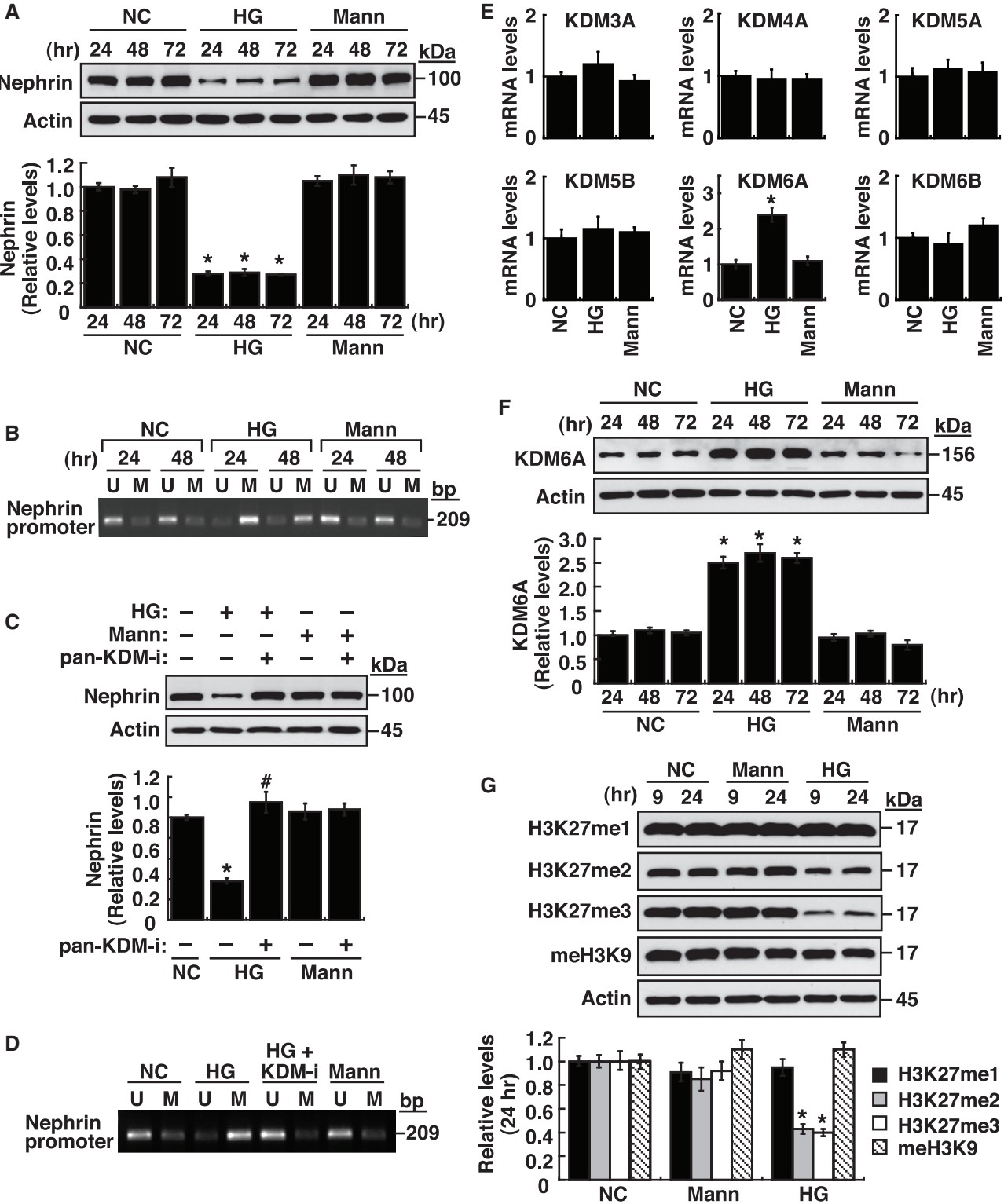

Figure 1.

including elevated levels of urinary protein excretion, relative kidney weights, and HbA1c were observed (Fig 3C and D). Importantly, based on the immunofluorescence analysis of kidney sections from normal mice and the 4-, 8- and 12-week diabetic mice

(Appendix Fig S2A), we evidently found that both increased KDM6A expression and reduced nephrin expression were closely associated with the presence of proteinuria in the 4-, 8-, and 12-week diabetic mice (Appendix Fig S2B). To further support the importance of

KDM6A in promoting diabetic kidney injury, GSK-J4 was applied to treat diabetic mice. Treatment with GSK-J4 significantly ameliorated diabetic urinary total protein release, kidney weight index, and urinary albumin leakage (Fig 3D and Appendix Fig S3A), as well as

attenuated diabetes-induced apoptosis of glomerular cells (TUNEL assay; Appendix Fig S3B). These results suggested that KDM6A plays a pivotal role in promoting diabetic kidney disease. We further found that GSK-J4 treatment restored levels of nephrin, WT1, and

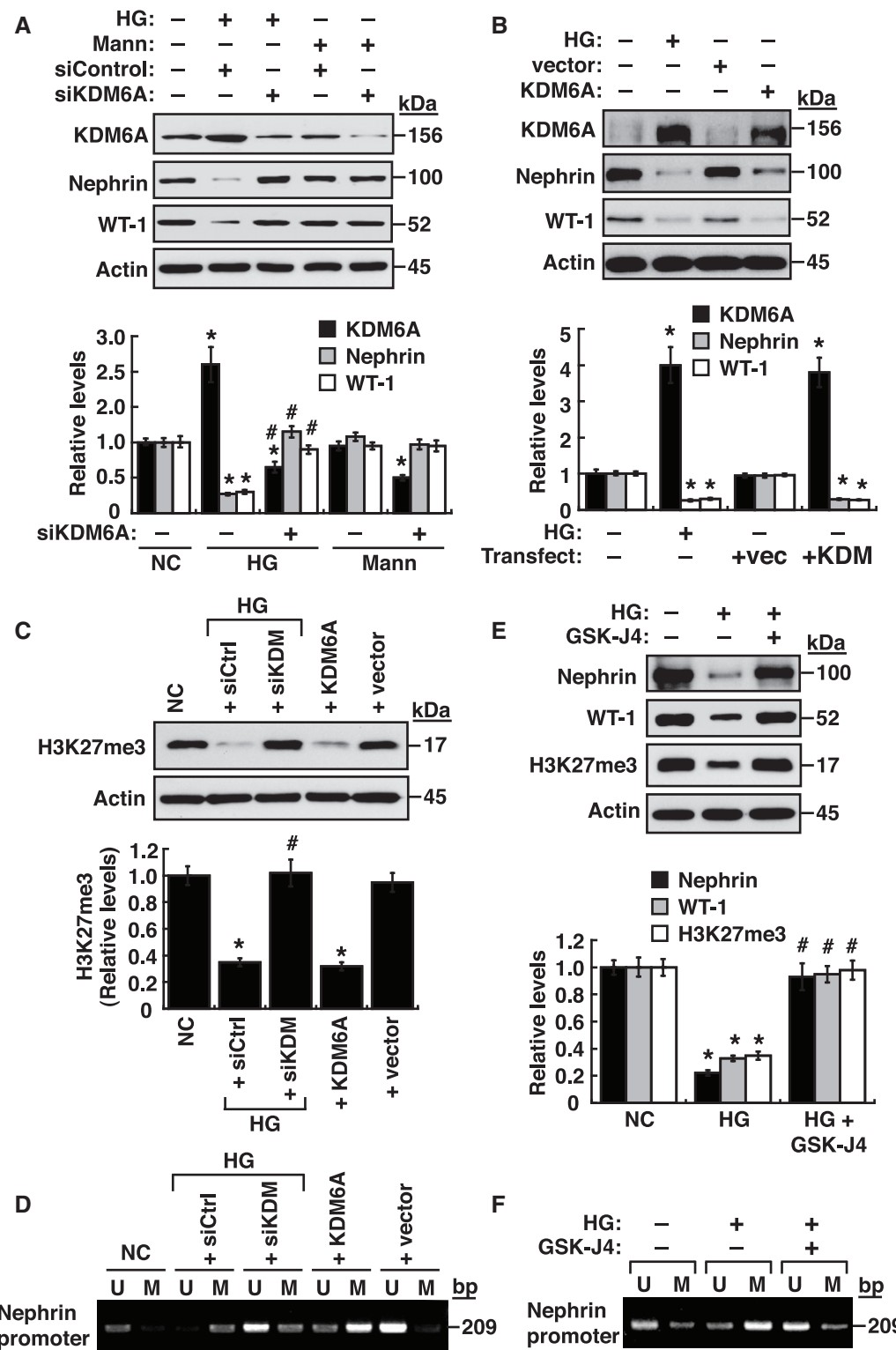

Figure 2.

Figure 2. KDM6A contributes to down-regulation of nephrin, WT-1, and H3K27me3 in podocytes.

A  Effect of KDM6A knockdown on the expression of nephrin and WT-1 in immortalized mouse podocytes cultured under high glucose conditions. Western blot analyses were performed using protein lysates of podocytes that were transfected with scrambled siRNAs or KDM6A-specific siRNAs, and cultured in high glucose (HG) or high mannitol (Mann) for 48 h. *P < 0.05 versus normal controls, #P < 0.05 versus control siRNA with HG incubation (parametric ANOVA and a Bonferroni *post hoc* test; n = 3).
B  Effect of KDM6A overexpression on expression levels of nephrin and WT-1. Immortalized podocytes were transfected with the KDM6A expression plasmid or the empty vector control for 48 h, and the expression levels of KDM6A, nephrin, and WT-1 in these transfected podocytes were analyzed by immunoblotting. *P < 0.05 versus vector controls (Parametric ANOVA and a Bonferroni *post hoc* test; n = 3).
C  Changes in H3K27me3 levels in podocytes after treatment with high glucose, the combination of high glucose and KDM6A knockdown, or KDM6A overexpression for 48 h. *P < 0.05 versus normal controls, #P < 0.05 versus control siRNA with HG incubation (Parametric ANOVA and a Bonferroni *post hoc* test; n = 3).
D  Changes in DNA methylation at nephrin gene promoter by high glucose, by the combination of high glucose plus KDM6A knockdown, or by KDM6A overexpression in immortalized podocytes. U, unmethylated-specific primers; M, methylated-specific primers. Experiments were repeated three times, and a representative gel from one experiment is shown.
E  Effect of GSK-J4 (40 μM), a specific KDM6A inhibitor, on high glucose-mediated changes in levels of nephrin, WT-1, and H3K27me3. *P < 0.05 versus normal controls, #P < 0.05 versus untreated HG-incubated cells (parametric ANOVA and a Bonferroni *post hoc* test; n = 3).
F  Changes in DNA methylation at nephrin gene promoter by GSK-J4 (40 μM) in immortalized podocytes. U, unmethylated-specific primers; M, methylated-specific primers. Experiments were repeated three times and a representative gel from one experiment is shown.

Data information: Data are expressed as mean ± SEM. See the exact *P*-values for comparison tests in Appendix Table S2.

H3K27me3 expression in STZ-induced diabetic kidney tissues (Fig 3E). Intriguingly, although GSK-J4 was considered to inhibit KDM6A demethylase activity (Kruidenier *et al*, 2012), we unexpectedly observed that GSK-J4 treatment also diminished KDM6A levels in diabetic kidney tissues as detected by immunoblotting (Fig 3E) or by immunofluorescence analysis (Fig 3F). These findings were supported further by examining the primary podocytes isolated from the above-treated mice (Fig 3G). The primary podocytes isolated from diabetic mice displayed abundant KDM6A expression and abnormal cell morphologies with the flattened, well-spread, and rounded cell shape (Herman-Edelstein *et al*, 2011), whereas podocytes isolated from GSK-J4-treated diabetic mice showed normal podocyte morphology with lower levels of KDM6A expression (Fig 3G). Since inactivation of KDM6A enzymatic activity by GSK-J4 in diabetic mice led to the reduction in KDM6A levels, we speculated that KDM6A might regulate its own expression through a positive feedback loop under diabetic conditions.

### Phenotypic characterization of podocyte-specific knockout of *KDM6A* in mice

To further study the *in vivo* contribution of KDM6A to podocyte dysfunction, we generated podocyte-specific *KDM6A*-knockout (*KDM6A*-KO) mice by crossing transgenic podocin-Cre mice with *KDM6*flox mice. In addition to *KDM6A* genotype analysis (Appendix Fig S4), the mRNA and protein expression of KDM6A in kidney glomeruli or podocytes isolated from *KDM6A*-KO mice were verified by quantitative RT–PCR and Western blot analysis, respectively (Fig 4A and B). All *KDM6A*-KO mice were viable and fertile under normal conditions. Following treatment with STZ, we found that there were no significant differences in body weights or blood glucose levels between wild-type and *KDM6A*-KO mice during the 8-week experimental period (Fig 4C). Consistently, there was also no difference in levels of HbA1c, a common marker for long-term glycemic control, between diabetic wild-type mice and diabetic *KDM6A*-KO mice (Fig 4D). However, proteinuria and kidney weights were significantly lower in diabetic *KDM6A*-KO mice than in diabetic wild-type mice (Figs 4D and EV1A–C), indicating that knockout of *KDM6A* in podocytes protected against diabetes-induced kidney injury. Additionally, as compared to diabetic

wild-type mice, we found that diabetic *KDM6A*-KO mice exhibited lower apoptotic cell numbers in glomeruli (TUNEL assay; Fig EV1D), lower ranges of glomerular basement membrane (GBM) thickness (electron microscopy; Fig EV1E), and less glomerular fibrosis (periodic acid–Schiff stain; Fig EV1F). Immunofluorescence analysis of kidney tissues also revealed that both nephrin and H3K27me3 were maintained at higher levels in diabetic *KDM6A*-KO mice relative to diabetic wild-type mice (Fig 4E and F). Furthermore, podocytes isolated from *KDM6A*-KO mice with diabetes retained normal morphological characteristics (Fig 4G) and exhibited normal levels of nephrin and WT-1 expression (Fig 4H). To further compare differences between wild-type and *KDM6A*-KO podocytes, the isolated primary podocytes were cultured in medium containing either normal or high glucose. We found that HG treatment decreased the expression levels of nephrin, WT-1 and podocin only in wild-type podocytes, but not in *KDM6A*-KO podocytes (Fig 4I). Since the transcription factor Snail was previously implicated in the repression of nephrin in podocytes (Matsui *et al*, 2007), the association between KDM6A and Snail in regulating nephrin expression was also examined (Fig. 4I). However, we did not find a correlation between Snail expression and KDM6A-mediated nephrin down-regulation in the experiments. Based on these results, we conclude that targeting KDM6A in podocytes could attenuate diabetes-induced kidney injury.

### Involvement of KLF10 in KDM6A-mediated podocyte dysfunction

To elucidate the mechanism by which KDM6A triggered podocyte dysfunction, we performed RNA sequencing (RNA-Seq) analysis of total RNAs extracted from primary podocytes that were infected with a control lentiviral vector or a lentiviral vector encoding KDM6A. From three independent experiments, up to 800 genes with more than twofold differential expression ($P < 0.05$) were found in KDM6A-overexpressing podocytes. Due to the complexity of these KDM6A-regulated genes, we focused our attention on transcriptional factors potentially involved in the control of nephrin gene expression (Beltcheva *et al*, 2003, 2010; Guo *et al*, 2004; Cohen *et al*, 2006; Matsui *et al*, 2007; Rascle *et al*, 2007; Ristola *et al*, 2012, 2013; Huang *et al*, 2013). From 86 selected genes (50 activators/repressors, 20 co-repressors, and 16 co-activators), only 6 genes (including *KLF10*, *KLF6*, *ETS1*, *KLF2*, *C/EBPα*, and *HDAC9*) were

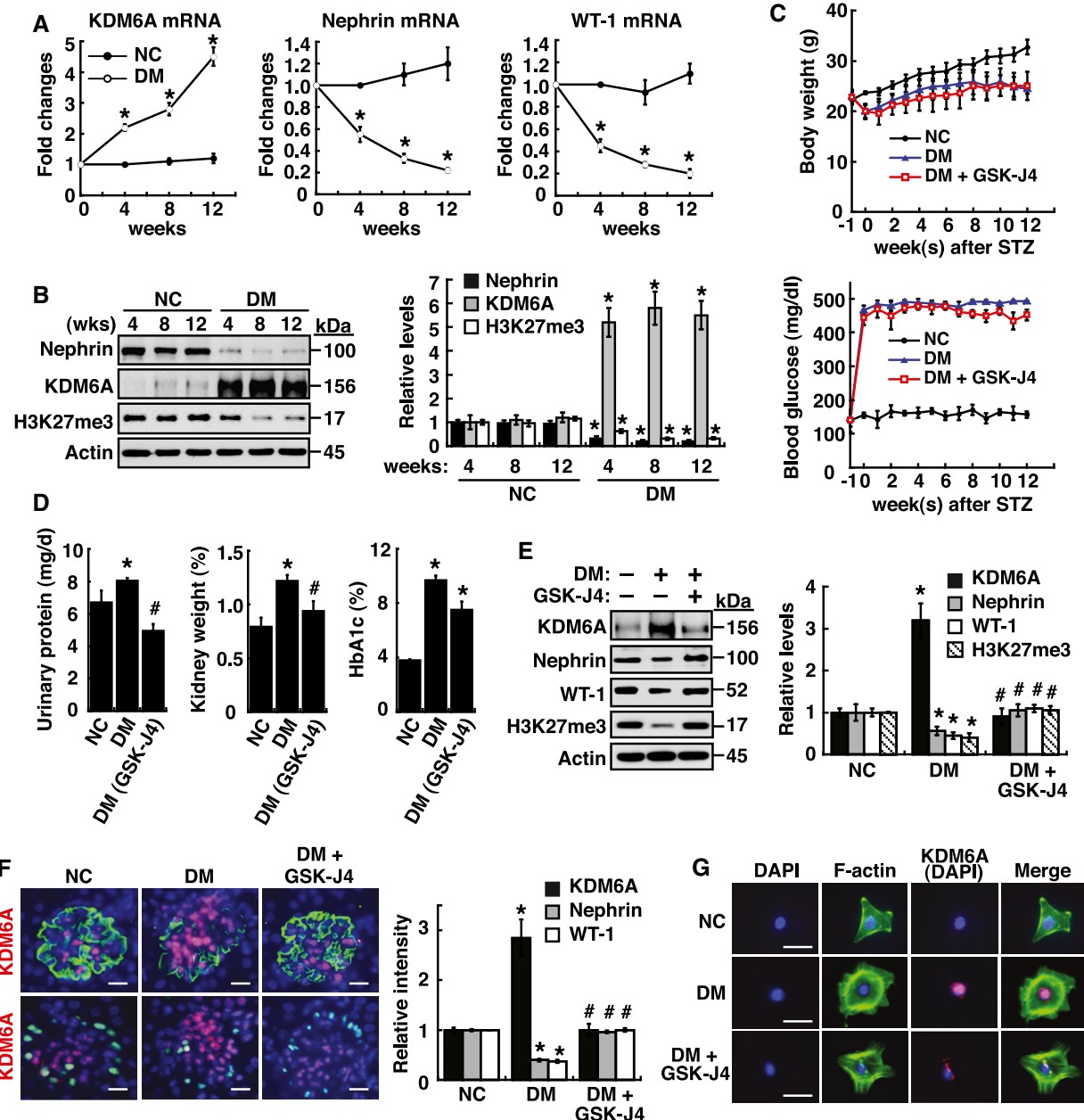

**Figure 3. KDM6A promotes podocyte and kidney dysfunction in diabetic mice.**

A  Relative mRNA levels of KDM6A, nephrin and WT-1, normalized to β-actin, expressed in kidney glomeruli of normal and diabetic mice at 4, 8, and 12 weeks after diabetic induction. *Significant differences ($P < 0.05$) compared with normal controls (Wilcoxon two-sample test; $n = 8$ each).

B  Western blot analysis of nephrin, KDM6A, and H3K27me3 in kidney glomeruli of normal and diabetic mice at different time points. *$P < 0.05$ versus normal controls for the indicated time points (Wilcoxon two-sample test; $n = 3$).

C  Changes in body weights and levels of blood glucose in normal, diabetic, and GSK-J4-treated diabetic mice. As noted, GSK-J4 treatment did not significantly affect body weights or blood glucose levels in diabetic mice during the experimental period of 12 weeks (Wilcoxon two-sample test; $n = 8$).

D  Levels of urinary protein excretion, weights of kidney, and levels of HbA1c in normal, diabetic, and GSK-J4-treated diabetic mice. Urinary total protein excretion, kidney weight, and levels of HbA1c were measured at 12 weeks after diabetic induction. The mean relative kidney weight (%) shown in the study is determined as the percent of kidneys out of total body weight, and the HbA1c level is defined as the ratio of HbA1c to the total hemoglobin (% HbA1c; DCCT unit). *$P < 0.05$ versus normal controls, #$P < 0.05$ versus untreated diabetic mice (parametric ANOVA and a Bonferroni *post hoc* test; $n = 8$).

E  Western blot analysis of KDM6A, nephrin, WT-1, and H3K27me3 expressed in kidney glomeruli of normal, diabetic, and GSK-J4-treated diabetic mice at 12 weeks after diabetic induction. *$P < 0.05$ versus normal controls, #$P < 0.05$ versus untreated diabetic mice (parametric ANOVA and a Bonferroni *post hoc* test; $n = 3$).

F  Immunofluorescence analysis of KDM6A, nephrin, and WT-1 in kidney sections from normal, diabetic, and GSK-J4-treated diabetic mice. Green: nephrin or WT-1; red: KDM6A; blue: DAPI. Scale bars, 20 μm. *$P < 0.05$ versus normal controls, #$P < 0.05$ versus untreated diabetic mice (parametric ANOVA and a Bonferroni *post hoc* test; $n = 3$).

G  Immunofluorescence staining of F-actin and KDM6A in primary podocytes isolated from normal, diabetic, and GST-J4-treated diabetic mice. Green: F-actin; red: KDM6A; blue: DAPI. Scale bars, 20 μm. Presented experiments were performed at least three times independently.

Data information: Data are expressed as mean ± SEM. See the exact *P*-values for comparison tests in Appendix Table S3.

identified with significant differences in responding to KDM6A over-expression (Fig 5A). Among these 6 transcription factors, KDM6A up-regulated *KLF10*, *KLF6*, and *ETS1*, but down-regulated *KLF2*,

*C/EBPα*, and *HDAC9* (Fig 5A). Notably, *KLF10* is the most up-regulated gene in the 86 selected genes and its association with kidney disease is not yet known. KLF10 was originally identified as a

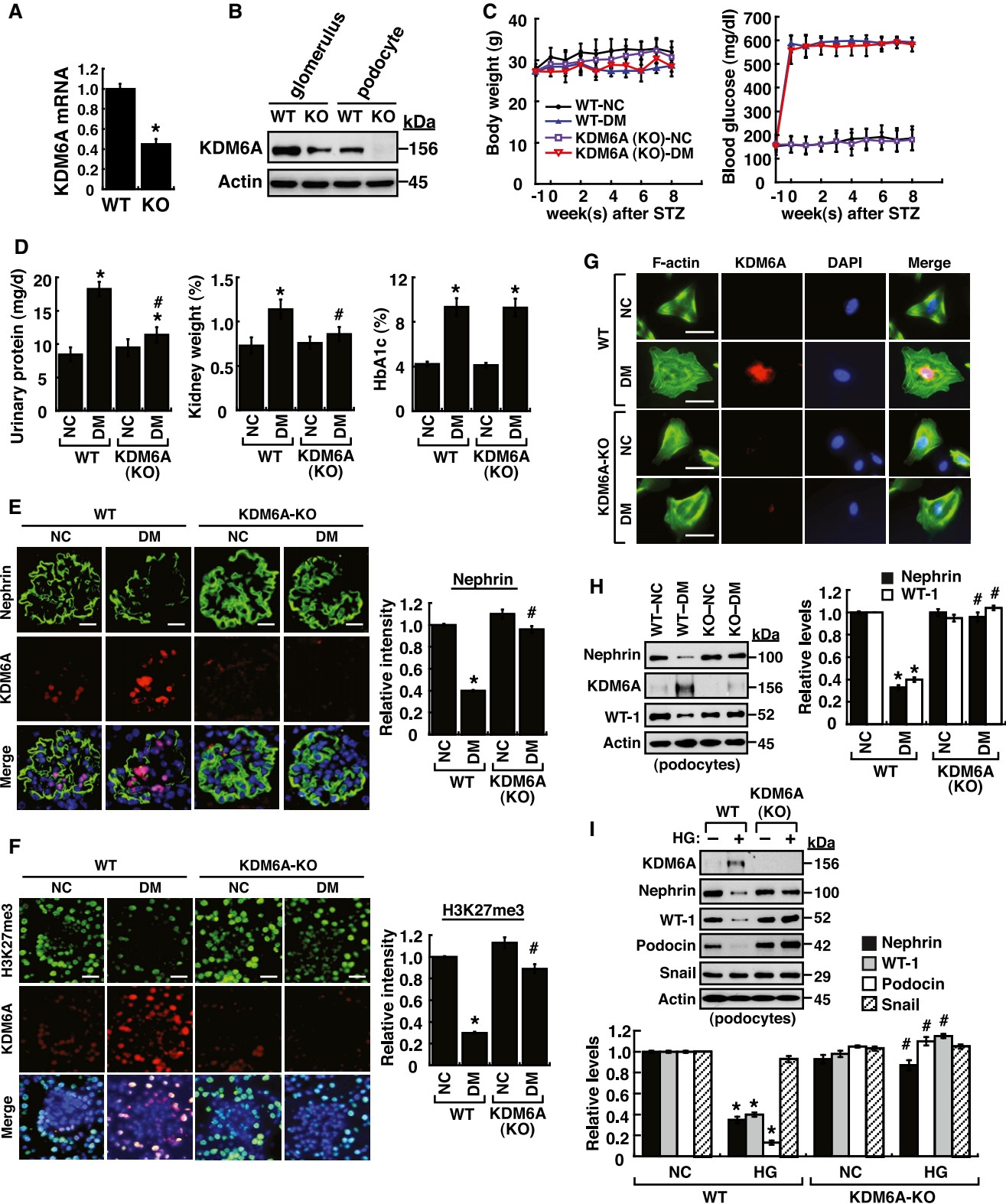

**Figure 4.**

**Figure 4. Podocyte-specific knockout of *KDM6A* in mice protects against diabetes-induced kidney injury.**

A Quantitative RT–PCR evaluation of *KDM6A* mRNA expression in kidney glomeruli isolated from wild-type and *KDM6A*-knockout (KO) mice. *P < 0.05, significant difference versus wild-type controls (Wilcoxon two-sample test; *n* = 8).

B Western blot analysis of KDM6A expression in glomeruli and podocytes isolated from wild-type and *KDM6A*-KO mice. Presented experiments were performed at least three times independently.

C Changes in body weights and levels of blood glucose in wild-type or *KDM6A*-KO mice with or without STZ treatment. No significant differences in body weights or blood glucose levels between wild-type and *KDM6A*-KO mice with diabetes were observed during the 8-week experimental period (Wilcoxon two-sample test; *n* = 8).

D Levels of urinary protein excretion, weights of kidney, and levels of HbA1c in wild-type and *KDM6A*-KO mice with or without STZ treatment (at 8 weeks after the onset of diabetes). The mean relative kidney weight (%) shown in the study is determined as the percent of kidneys out of total body weight, and the HbA1c level is defined as the ratio of HbA1c to the total hemoglobin (% HbA1c; DCCT unit). *P < 0.05 versus untreated wild-type controls, #P < 0.05 versus STZ-treated wild-type mice (parametric ANOVA and a Bonferroni *post hoc* test; *n* = 8).

E Immunofluorescence images of kidney sections stained with KDM6A and nephrin in wild-type (WT-NC), *KDM6A*-KO (KO-NC), STZ-treated wild-type (WT-DM), or STZ-treated *KDM6A*-KO (KO-DM) mice. Scale bars, 20 μm. *P < 0.05 versus untreated wild-type controls, #P < 0.05 versus STZ-treated wild-type mice (parametric ANOVA and a Bonferroni *post hoc* test; *n* = 3).

F Immunofluorescence images of kidney sections stained with KDM6A and H3K27me3 in wild-type (WT-NC), *KDM6A*-KO (KO-NC), STZ-treated wild-type (WT-DM), or STZ-treated *KDM6A*-KO (KO-DM) mice. Scale bars, 20 μm. *P < 0.05 versus untreated wild-type controls, #P < 0.05 versus STZ-treated wild-type mice (parametric ANOVA and a Bonferroni *post hoc* test; *n* = 3).

G Immunofluorescence staining of F-actin and KDM6A in primary podocytes isolated from the above-treated mice. Presented experiments were performed at least three times independently. Scale bars, 20 μm.

H Western blot analysis of nephrin, KDM6A, and WT-1 expressed in primary podocytes isolated from the above-treated mice. *P < 0.05 versus untreated wild-type controls, #P < 0.05 versus STZ-treated wild-type mice (parametric ANOVA and a Bonferroni *post hoc* test; *n* = 3).

I Effect of high glucose on the expression of podocyte-related markers in primary cultured podocytes isolated from wild-type or *KDM6A*-KO mice. The primary podocytes isolated from wild-type or KDM6A-KO mice were cultured in normal or high glucose (30 mM) for 48 h. Protein lysates from the cultured podocytes were subjected to Western blot analysis with the indicated antibodies. *P < 0.05 versus wild-type podocytes in normal glucose, #P < 0.05 versus wild-type podocytes in high glucose (parametric ANOVA and a Bonferroni *post hoc* test; *n* = 3).

Data information: Data are expressed as mean ± SEM. See the exact *P*-values for comparison tests in Appendix Table S4.

TGF-β-inducible early gene 1 (TIEG) in human osteoblasts and often functions as a transcriptional repressor involved in multiple cellular processes (Subramaniam *et al*, 1995, 2010; McConnell & Yang, 2010; Yang *et al*, 2017). We therefore decided to further investigate the role of KLF10 in podocyte injury. Increased KLF10 expression was confirmed in primary cultured podocytes with KDM6A overexpression or with HG treatment (Fig 5B and C). Importantly, knockdown of KLF10 in primary podocytes significantly inhibited HG-mediated reduction of nephrin (Fig 5D), indicating that KLF10 acts as a negative regulator of nephrin expression. Surprisingly, we noticed that KLF10 knockdown also resulted in the reduction of KDM6A expression in HG-treated podocytes (Fig 5D).

In our mouse models, increased KLF10 expression along with reduced nephrin expression was observed by immunofluorescence analysis in diabetic kidney tissues (Fig 5E and F). Particularly, pharmacological inhibition or podocyte-specific knockout of KDM6A substantially prevented KLF10 up-regulation in diabetic kidney tissues (Fig 5E and F), supporting that KLF10 is a downstream effector of KDM6A. Similar results were also obtained when KLF10 expression was determined by immunoblotting using primary podocytes isolated from the above-treated mice (Fig 5G and H). To investigate whether KLF10 directly modulated nephrin gene expression, we performed electrophoretic mobility shift assays (EMSAs), supershift assays, and chromatin immunoprecipitation (ChIP) on nephrin gene promoter (Figs 5I and EV2). In mouse nephrin gene promoter, there are two evolutionarily conserved regions located from −188 to −270 (83-bp; WT-1 binding) and from −1,870 to −2,106 (237-bp; Sp1 binding), which are essential for podocyte-specific expression (Guo *et al*, 2004; Beltcheva *et al*, 2010). Electrophoretic mobility shift assay experiments showed that KLF10 directly bound to the promoter element from −1,893 to −1,931 (Fig EV2), a region previously identified as an SP-1 binding site in the nephrin promoter (Beltcheva *et al*, 2010). ChIP assays also demonstrated enhanced binding of KLF10 to the nephrin promoter region from −2,052 to −1,753 or from −1,802 to −1,553, along with Dnmt1 (but not Dnmt3), in primary podocytes cultured in HG (Fig 5I). When KLF10 was overexpressed in primary cultured podocytes, we observed that all tested podocyte-specific markers, including nephrin, WT1, podocin, and synaptopodin, were repressed (Fig 5J); however, KDM6A expression could be conversely activated (Fig 5J). These results strongly suggested that KLF10 could have multiple actions in promoting podocyte dysfunction.

## Protection from diabetes-induced podocyte injury in *KLF10*-knockout mice

To further study the function of KLF10 in kidney disease, we used *KLF10*-KO mice as described previously (Yang *et al*, 2013). Expression of KLF10 in kidney tissues of *KLF10*-KO mice was confirmed in our laboratory (Fig 6A and B). Following treatment with STZ, we found that KLF10 depletion significantly protected mice against diabetes-induced proteinuria (urinary excretion of total protein and albumin), glomerular cell apoptosis, GBM thickening, and even renal fibrosis (Figs 6C and D, and EV3). This was further supported by the findings that nephrin expression in kidney tissues did not significantly change between the STZ-treated *KLF10*-KO mice and the untreated wild-type or untreated *KLF10*-KO mice (Fig 6E). Intriguingly, when the expression levels of KDM6A were examined, we found that diabetic induction could not increase KDM6A expression in kidney tissues or primary podocytes isolated from *KLF10*-KO mice as determined by immunofluorescence or by immunoblotting (Fig 6F–I). These results supported that KLF10 might conversely regulate KDM6A expression *in vivo* under diabetic conditions. Since *KLF10* is a key TGF-β-inducible early growth response gene, we treated podocytes with TGF-β1 to further study the relationship between KLF10 and KDM6A. After treatment with TGF-β1, both

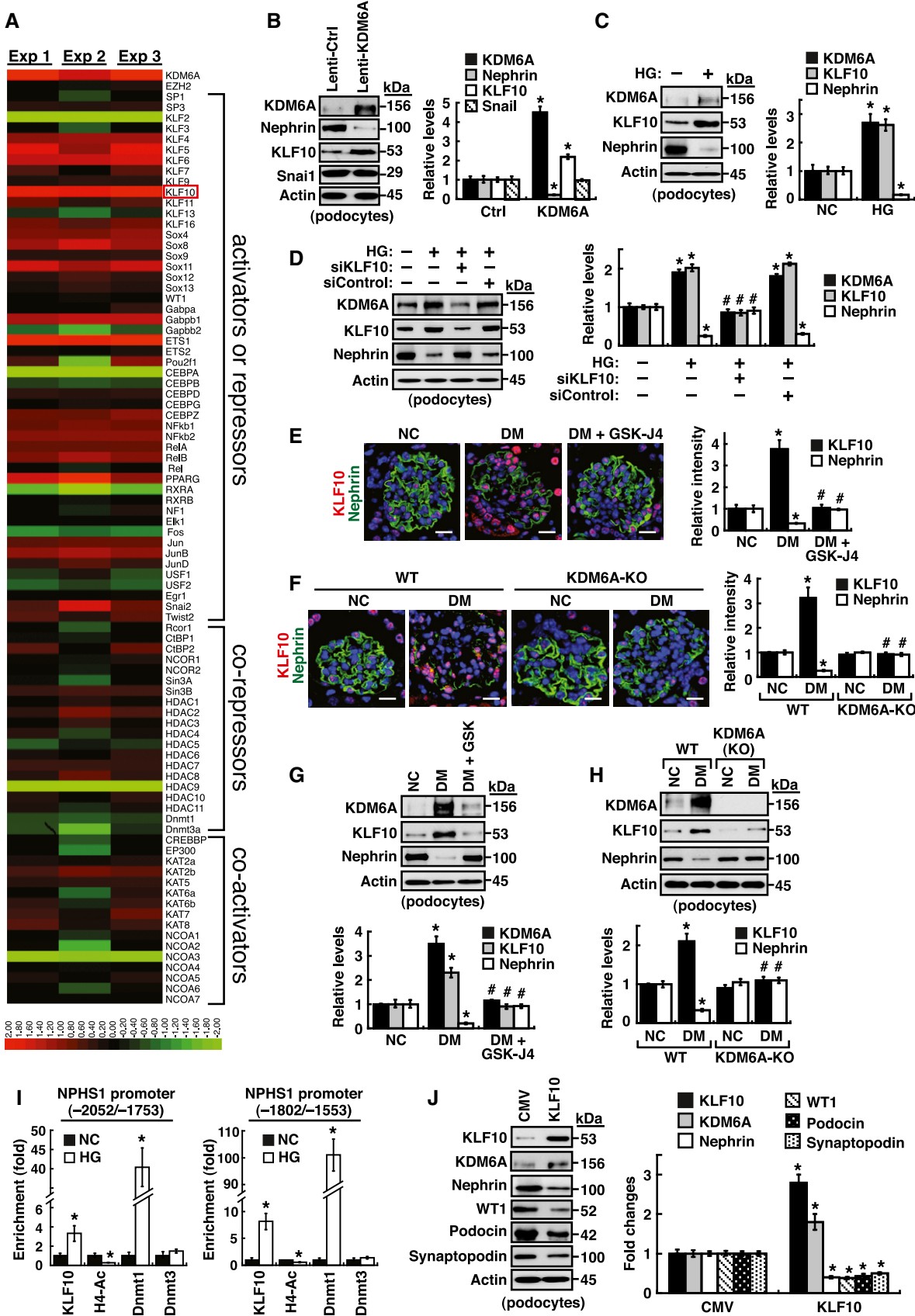

**Figure 5.**

**Figure 5.  KLF10 is a downstream effector of KDM6A and is capable of directly binding to nephrin gene promoter.**

A   Screening of the potential KDM6A-regulated transcriptional factors involved in repression of nephrin expression. After transduction with a control lentiviral vector or a lentiviral vector expressing KDM6A into primary podocytes for 48 h, intracellular RNAs were isolated and used for RNA sequencing (RNA-Seq) analysis. The transcript expression patterns of 86 selected transcriptional factors from three independent RNA-Seq experiments are presented in a heat map. Notably, among these 86 selected genes, *KLF10* is the most up-regulated gene.

B   Validation of increased KDM6A and KLF10 expression in primary podocytes that were infected with an empty lentiviral vector or KDM6A-expressing lentiviral vector for 48 h. *$P < 0.05$ versus the empty vector control (Wilcoxon two-sample test; $n = 3$).

C   Increased expression of KDM6A and KLF10 in primary podocytes cultured in high glucose for 48 h. *$P < 0.05$ versus normal controls (Wilcoxon two-sample test; $n = 3$).

D   Effect of KLF10 knockdown on high glucose-mediated reduction of nephrin in primary podocytes. As noted, knockdown of KLF10 prevented down-regulation of nephrin and up-regulation of KDM6A in high glucose-treated podocytes. *$P < 0.05$ versus normal controls, #$P < 0.05$ versus control siRNA with HG incubation (parametric ANOVA and a Bonferroni *post hoc* test; $n = 3$).

E   Immunofluorescence analysis of KLF10 and nephrin in renal sections of normal, diabetic, and GSK-J4-treated diabetic mice. Scale bars, 20 μm. *$P < 0.05$ versus the normal control group, #$P < 0.05$ versus the untreated diabetic group (parametric ANOVA and a Bonferroni *post hoc* test; $n = 3$).

F   Immunofluorescence images of KLF10 and nephrin in renal sections of wild-type or *KDM6A*-KO mice with or without STZ treatment. Scale bars, 20 μm. *$P < 0.05$ versus the untreated wild-type group, #$P < 0.05$ versus the STZ-treated wild-type group (parametric ANOVA and a Bonferroni *post hoc* test; $n = 3$).

G   Western blot analysis of KDM6A, KFL10, and nephrin expression in primary podocytes isolated from normal, diabetic, and GSK-J4-treated diabetic mice. *$P < 0.05$ versus normal controls, #$P < 0.05$ versus the untreated diabetic group (parametric ANOVA and a Bonferroni *post hoc* test; $n = 3$).

H   Western blot analysis of KDM6A, KLF10, and nephrin expression in primary podocytes isolated from wild-type or *KDM6A*-KO mice with or without STZ treatment. *$P < 0.05$ versus the untreated wild-type group, #$P < 0.05$ versus the STZ-treated wild-type group (parametric ANOVA and a Bonferroni *post hoc* test; $n = 3$).

I    ChIP analysis of KLF10, acetyl-histone H4 (H4-Ac), Dnmt1 and Dnmt3 binding to nephrin gene promoter. ChIP assays were carried out using cross-linked chromatin from primary podocytes that were cultured in normal or high glucose conditions. *$P < 0.05$, significant difference versus the normal control group (Wilcoxon two-sample test; $n = 3$).

J    Modulation of podocyte-specific marker expression in primary podocytes by KLF10 overexpression. Ectopic overexpression of KLF10 in primary podocytes significantly repressed various podocyte-specific markers, but conversely increased KDM6A expression. *$P < 0.05$ versus the empty vector control (Wilcoxon two-sample test; $n = 3$).

Data information: Data are expressed as mean ± SEM. See the exact *P*-values for comparison tests in Appendix Table S5.

KLF10 and KDM6A could be activated simultaneously, along with reduced expression of nephrin and WT-1 (Fig 6J). However, when KLF10 was knocked down in TGF-β1-treated podocytes, we noticed that KDM6A expression was also concomitantly reduced (Fig 6J). Taken together, our findings suggested KDM6A and KLF10 positively regulate each other's expression in podocytes under stress conditions (Fig 6K).

To further determine the interlink between KDM6A and KLF10 in mediating podocyte dysfunction, immortalized mouse podocytes overexpressing KDM6A in combination with KLF10 knockdown (Fig EV4A) or podocytes overexpressing KLF10 in combination with KDM6A knockdown (Fig EV4B) were subjected to Western blot analysis for the expression levels of nephrin and WT-1. In these experiments, we found that down-regulation of nephrin and WT-1 was closely correlated with increased KLF10, but not increased KDM6A (Fig EV4A and B), indicating that KLF10 plays a dominant role in regulating podocyte-specific marker proteins. Additionally, due to the fact that TGF-β1 is a multifunctional cytokine critically involved in podocyte dysfunction, the potential action of TGF-β1 in the inter-regulation between KDM6A and KLF10 was investigated (Fig EV5). We here showed that inhibition of TGF-β1 with a neutralizing antibody (10 μg/ml) was able to block HG-mediated up-regulation of KDM6A and KLF10, and down-regulation of nephrin and WT-1 in podocytes (Fig EV5A). However, addition of the same amount of anti-TGF-β1 neutralizing antibody in KDM6A- or KLF10-overexpressing podocytes could not influence the expression of their downstream targets including nephrin, WT-1 and KLF10 or KDM6A (Fig EV5B and C). These results indicated that although TGF-β1 is an important stimulator for the KDM6A–KLF10 positive feedback loop, it is not essential for mediating the positive inter-regulation between KDM6A and KLF10 (Fig EV5D).

## Increased levels of KDM6 and KLF10 in kidney tissues or urinary exosomes of human patients with diabetic nephropathy

When human nephrectomy tissue samples from patients with or without diabetic nephropathy ($n = 6$ for each group) were examined by immunofluorescence or by immunoblotting, we consistently found that increased levels of KDM6A and KLF10, accompanied by reduced levels of nephrin and WT-1, were significantly detected in kidney tissues of diabetic nephropathy patients as compared to control subjects (Fig 7A and B). Moreover, to further confirm our study findings, we attempted to examine levels of KDM6A, KLF10, and nephrin mRNAs in urinary exosomes of control subjects and patients with diabetic nephropathy ($n = 12$ for each group). The baseline characteristics of study participants are shown in Appendix Table S12. Quantitative analysis of mRNA contents in human urinary exosomes also showed that higher levels of exosomal KDM6A and KLF10 mRNAs, accompanied by lower levels of nephrin mRNA, were significantly observed in diabetic nephropathy subjects than in control subjects (Fig 7C).

## Discussion

Despite extensive efforts, there are not enough therapeutic interventions in preventing the progression of diabetic nephropathy, suggesting that additional detrimental pathways for diabetic nephropathy need to be further explored. In this report, we present a reinforcing feedback loop that involves KDM6A and KLF10 in podocytes, critically promoting podocyte dysfunction under diabetic conditions. Inactivation of KDM6A and KLF10 in mice can significantly protect against diabetes-induced proteinuria and kidney injury.

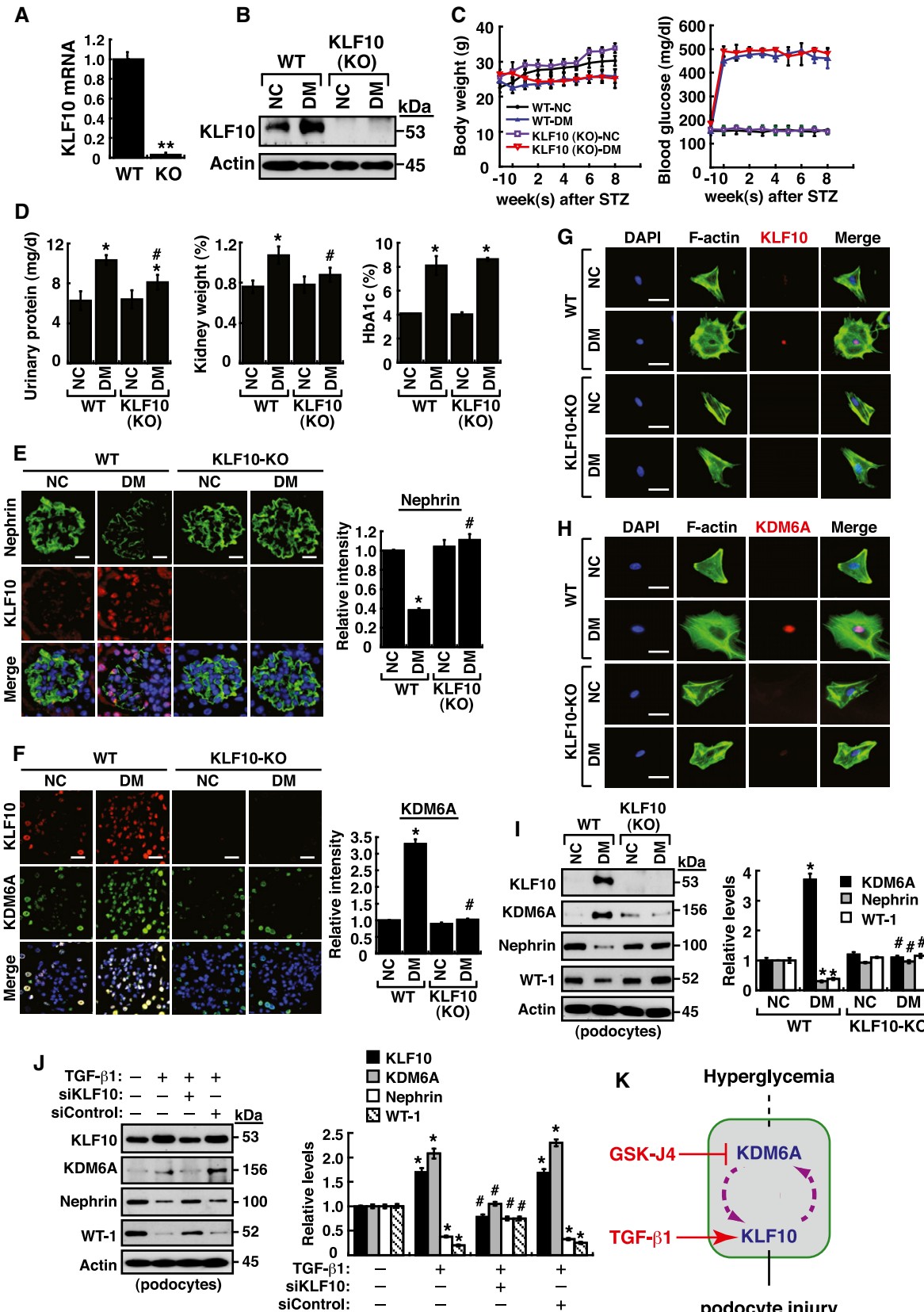

Figure 6.

◀

**Figure 6. KDM6A and KLF10 can build a positive feedback loop that is critically involved in podocyte dysfunction.**

A    Quantitative RT–PCR evaluation of *KLF10* mRNA expression in kidney samples of wild-type and *KLF10*-knockout (KO) mice. **$P < 0.0001$, significant difference versus wild-type controls (Wilcoxon two-sample test; $n = 8$).

B    Western blot analysis of KLF10 expression in kidney samples of wild-type or *KLF10*-KO mice treated with or without STZ at 8 weeks after the onset of diabetes. Presented experiments were performed at least three times independently.

C    Changes in body weight and blood glucose levels in wild-type or *KLF10*-KO mice with or without STZ treatment. No significant differences in body weights or blood glucose levels between diabetic wild-type mice and diabetic *KLF10*-KO mice were detected (Wilcoxon two-sample test; $n = 8$).

D    Levels of urinary protein excretion, weights of kidney, and levels of HbA1c in wild-type or *KLF10*-KO mice with or without STZ treatment. The mean relative kidney weight (%) shown in the study is determined as the percent of kidneys out of total body weight, and the HbA1c level is defined as the ratio of HbA1c to the total hemoglobin (% HbA1c; DCCT unit). *$P < 0.05$ versus untreated wild-type controls, #$P < 0.05$ versus STZ-treated wild-type mice (parametric ANOVA and a Bonferroni *post hoc* test; $n = 8$).

E    Immunofluorescence images of kidney sections stained with KLF10 and nephrin in wild-type or *KLF10*-KO mice untreated or treated with STZ. Scale bars, 20 μm. *$P < 0.05$ versus untreated wild-type controls, #$P < 0.05$ versus STZ-treated wild-type mice (parametric ANOVA and a Bonferroni *post hoc* test; $n = 3$).

F    Immunofluorescence images of kidney sections stained with KLF10 and KDM6A in wild-type or *KLF10*-KO mice untreated or treated with STZ. Scale bars, 20 μm. *$P < 0.05$ versus untreated wild-type controls, #$P < 0.05$ versus STZ-treated wild-type mice (parametric ANOVA and a Bonferroni *post hoc* test; $n = 3$).

G    Immunofluorescence staining of F-actin and KLF10 in primary podocytes isolated from the wild-type or *KLF10*-KO mice untreated or treated with STZ. Presented experiments were performed at least three times independently. Scale bars, 20 μm.

H    Immunofluorescence staining of F-actin and KDM6A in primary podocytes isolated from the wild-type or *KLF10*-KO mice untreated or treated with STZ. Presented experiments were performed at least three times independently. Scale bars, 20 μm.

I    Western blot analysis of KLF10, KDM6A, nephrin, and WT-1 expression in primary podocytes isolated from the wild-type or *KLF10*-KO mice that were untreated or treated with STZ. *$P < 0.05$ versus untreated wild-type controls, #$P < 0.05$ versus STZ-treated wild-type mice (parametric ANOVA and a Bonferroni *post hoc* test; $n = 3$).

J    Western blot analysis of KLF10, KDM6A, nephrin, and WT-1 expressed in podocytes treated with TGF-β1 or the combination of TGF-β1 and KLF10 knockdown. *$P < 0.05$ versus normal controls, #$P < 0.05$ versus control siRNA with TGF-β1 treatment (parametric ANOVA and a Bonferroni *post hoc* test; $n = 3$).

K    A model for the reciprocal regulation between KDM6A and KLF10 in podocytes under diabetic conditions. Under hyperglycemic conditions, increased KDM6A or KLF10 may regulate each other and create a positive feedback loop that aggravates podocyte dysfunction.

Data information: Data are expressed as mean ± SEM. See the exact *P*-values for comparison tests in Appendix Table S6.

---

According to the findings from the Diabetes Control and Complication Trial (DCCT) and Epidemiology of Diabetes Intervention and Complications (EDIC) studies (Colagiuri *et al*, 2002; Writing Team for the Diabetes *et al*, 2003; Nathan *et al*, 2005), diabetic nephropathy can often persist and continue to enter end-stage renal disease in diabetic patients even with intensive glucose control. The occurrence of a "metabolic memory" phenomenon in kidney cells may be attributed in part to epigenetic regulation. In our studies, we found that elevated KDM6A levels and reduced H3K27me3 levels, together with a significant decrease in nephrin, were consistently detected in podocytes *in vitro* and *in vivo* under hyperglycemic conditions. Our findings are generally consistent with the previous study by Majumder *et al* (2018), which showed the involvement of KDM6A in podocyte dedifferentiation. Besides, we here provide at least four additional pieces of information to further support the importance of KDM6A in promoting podocyte dysfunction. First, podocyte-specific knockout of *KDM6A* in mice substantially protects against diabetes-induced proteinuria and kidney injury. Second, knockout of *KDM6A* in podocytes prevents HG-induced podocyte dysfunction *in vitro*. Third, increased KDM6A in podocytes sufficiently suppresses numerous podocyte differentiation markers and promotes a positive feedback loop to maintain its high-level expression under diabetic conditions. Fourth, elevated levels of KDM6A protein or mRNA could be also observed in kidney tissues or in urinary exosomes of human diabetic nephropathy patients as compared to controls.

In addition to the H3K37me2/3 demethylating function, KDM6A has been proposed to have demethylase-independent functions in the regulation of gene expression through interacting with other transcriptional regulators (Van der Meulen *et al*, 2014; Ezponda *et al*, 2017). KDM6A can be a component of MLL2 complex, a histone H3K4 methyltransferase complex (Van der Meulen *et al*,

2014). The dynamic interplay of H3K27me2/3 demethylation and H3K4 methylation, in couple with chromatin remodeling by the BRG1-containing SWI/SNF complex and with H3K27 acetylation by HATs (e.g., CBP), may profoundly promote chromatin relaxation and activate specific gene activation. Despite the possible interactions between KDM6A and various histone-modifying enzymes, we currently do not know whether the demethylase-independent action of KDM6A plays a role in the regulation of KLF10 or other downstream targets that are potentially involved in podocyte dysfunction. Notably, Majumder *et al* (2018) have also presented that loss of the repressive H3K27me3 mark in podocytes either by down-regulation of EZH2 or by overexpression of KDM6A triggered activation of the Notch ligand Jag1 and the subsequent Notch signaling pathway. However, in our RNA-Seq analysis, we did not detect significant changes in mRNA levels of Jag1 and other Notch receptors (fold change with a cutoff of 2, $P < 0.05$) in podocytes overexpressing KDM6A. These results suggest that some subtle changes may be not evidently discernible in the RNA-Seq experiments and more research and validation studies are required for determining these subtle changes in podocytes.

To find out which of the expression changes in RNA-Seq experiments are the most important causes of podocyte dysfunction, we focused our attention on transcriptional factors involved in the regulation of nephrin expression and found that KLF10 is a candidate target. There are currently a total of 27 SP/KLF family members identified in mouse or human, which include nine members of the SP subfamily (SP1–SP9) and 18 members of the KLF subfamily (KLF1–KLF18; McConnell & Yang, 2010; Bialkowska *et al*, 2017; Memon & Lee, 2018). Due to the structural similarities in the SP/KLF DNA-binding domains at their C-terminal regions and the diverse functional (transactivation/repression) domains at their N-terminal regions, the SP/KLF members may have overlapping sets

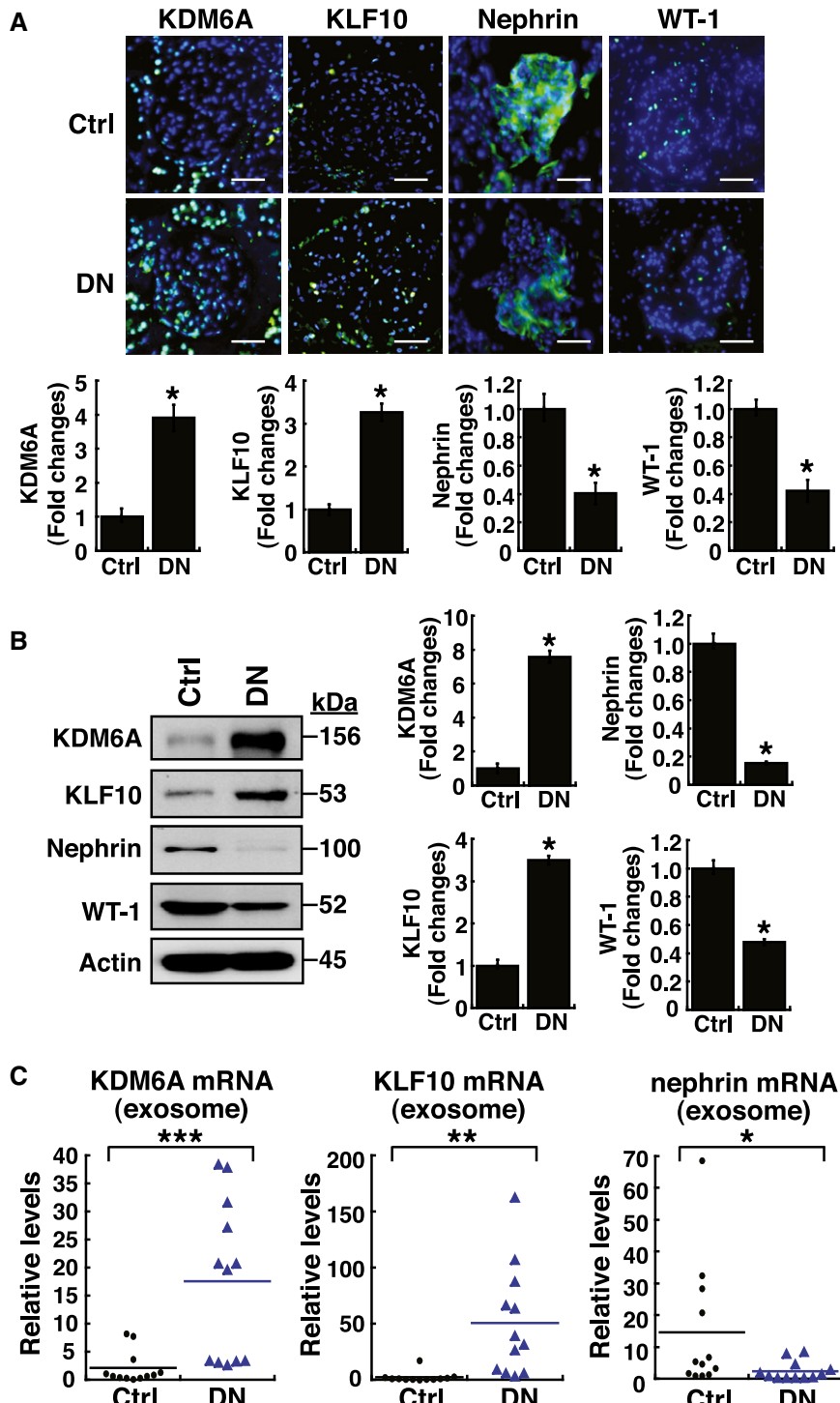

**Figure 7. Elevated levels of KDM6A and KLF10 expression, along with decreased levels of nephrin (or WT-1) expression, are detected in kidney tissues and urinary exosomes of patients with diabetic nephropathy.**

A  Immunofluorescence analysis of kidney sections from diabetic nephropathy subjects and non-diabetic controls stained with KDM6A, KLF10, nephrin, and WT-1. Scale bars, 50 μm. *$P < 0.05$ by Wilcoxon two-sample test ($n = 6$ for each group).

B  Western blot analysis of kidney tissues from non-diabetic controls and diabetic nephropathy subjects. Expression levels of KDM6A, KLF10, nephrin, and WT-1 in kidney tissues were determined by immunoblotting using the indicated antibodies. Relative protein levels in kidney tissues were normalized to actin. *$P < 0.05$ by Wilcoxon two-sample test ($n = 6$ for each group).

C  Levels of KDM6A and KLF10, nephrin mRNAs in urine exosomes of diabetic nephropathy patients and non-diabetic controls. Relative mRNA levels in human urinary exosomes were normalized to 18S rRNA. Horizontal lines are medians. *$P < 0.05$, **$P < 0.01$, and ***$P < 0.001$ by Wilcoxon two-sample test ($n = 12$ for each group).

Data information: Data are expressed as mean ± SEM (A and B). See the exact *P*-values for comparison tests in Appendix Table S7.

of target genes but display diverse influences on target gene expression. By comparing the expression profiling of all 17 KLFs (KLF18 not included) in three RNA-Seq datasets, we found that KLF members including KLF1, KLF8, KLF12, KLF14, KLF15, and KLF17 were expressed at very low (or undetectable) levels in podocytes with or without KDM6A overexpression. Among the other KLFs, KLF10 is the most significantly up-regulated gene in response to KDM6A overexpression. KLF10 was originally identified as a TGF-β-inducible early gene product in human osteoblast cells (Subramaniam *et al*, 1995). In addition to TGF-β, KLF10 could be induced by bone morphogenetic proteins (BMPs), estrogen, nerve growth factor, and epidermal growth factor (EGF) depending on cell types and environmental conditions (Subramaniam *et al*, 2010; Memon & Lee, 2018). Although several KLF members including KLF2, KLF4, KLF6, and KLF15 in glomerular endothelial cells or podocytes have been linked to kidney diseases, all these reported KLF members serve as renal protectors (Mallipattu *et al*, 2012, 2015, 2017; Hayashi *et al*, 2014, 2015; Zhong *et al*, 2015, 2016). To date, KLF10 is the only SP/KLF member that endangers podocyte function.

Multiple actions of KLF10 in podocyte dysfunction or injury can be proposed. First, KLF10 can directly bind nephrin gene promoter along with the recruitment of Dnmt1 to repress nephrin expression. Particularly, our results revealed that one of KLF10-binding sites in the nephrin gene promoter is known as a previously identified SP1-binding site located in the evolutionarily conserved promoter region. Notably, studies by Hayashi *et al* (2014, 2015) have previously shown that mutually exclusive binding of KLF4 (a transcriptional activator) and Dnmt1 to the nephrin gene promoter resulted in activation and repression of nephrin expression, respectively. Based on these previous studies and our present study, we here propose that SP1 and KLF4 act as transcriptional activators of the nephrin gene promoter, whereas KLF10 and Dnmt1 function as transcriptional repressors of the nephrin gene promoter. Second, in addition to the repression of nephrin expression, overexpression of KLF10 can also suppress other podocyte-specific genes such as WT1, podocin, and synaptopodin in podocytes. Currently, it remains unclear whether the repression of these genes by KLF10 is mediated through direct binding of KLF10 to these gene promoters. Third, many studies have shown that KLF10 overexpression mimics anti-proliferative effect of TGF-β and induces cell apoptosis in various cell types (Ribeiro *et al*, 1999; Johnsen *et al*, 2002; Jin *et al*, 2007). Therefore, it could be possible that induction of podocyte dysfunction by KLF10 may be partly due to its anti-proliferative and pro-apoptotic activities. Fourth, KLF10 can conversely activate KDM6A, thereby promoting global epigenetic reprogramming and leading to aberrant gene expression. Intriguingly, although previous studies have reported that some abnormalities such as in the microarchitecture and mechanical properties of tendons could be observed in KLF10-deficient mice (Bensamoun *et al*, 2006a,b; Haddad *et al*, 2009; Gumez *et al*, 2010), we here found that *KLF10*-KO mice actually had a normal lifespan and displayed grossly normal phenotype relative to wild-type mice under basal conditions. These findings suggest that KLF10 can be an excellent therapeutic target in the treatment of diabetic nephropathy because its down-regulation may not cause serious long-term side effects.

In conclusion, our study presents a new deleterious positive feedback regulation that involves KDM6A and KLF10 in podocytes

under diabetic conditions, which may facilitate the induction of irreversible kidney injury. Moreover, in contrast to the previously reported KLFs that act to protect kidney from damage, we here show that KLF10 has a unique role in promoting kidney dysfunction. These findings may open avenues for therapeutic manipulation in attenuating diabetes-induced kidney injury.

## Materials and Methods

### Cell cultures, reagents, and transfections

Conditionally immortalized mouse podocyte cell line (Mundel *et al*, 1997), which harbors a thermosensitive variant of SV40-T antigen, was growth in RPMI-1640 medium containing 10% fetal bovine serum (FBS) and interferon-gamma (50 U/ml) at 33°C as described previously (Lin *et al*, 2014). To stimulate cell differentiation, the culture temperature was shifted from 33 to 37°C for 8 days. Primary podocytes isolated from mice were cultured in RPMI-1640 supplemented with 10% FBS. To mimic a hyperglycemic condition, immortalized mouse podocytes or primary mouse podocytes ($5 \times 10^5$ cells/well, 6-well plate) were cultured in high glucose (30 mM), and mannitol served as an osmotic control for high glucose. Additionally, 500 or 1,000 nM of pargyline hydrochloride (P8013; Sigma), 40 μM of GSK-J4 (4594; Tocris Bioscience, UK), and 5 ng/ml of TGF-β1 (PeproTech, Rocky Hill, NJ) were used for all indicated experiments. Transfection experiments were carried out using Lipofectamine 2000 reagent according to the manufacturer's instruction (Thermo Fisher Scientific).

### Methylation-specific PCR

Genomic DNA was isolated from cultured podocytes using DNeasy Blood & Tissue kit (Qiagen Hilden, Germany). One microgram of genomic DNA was bisulfite converted using EpiTect Fast Bisulfite Conversion kit (Qiagen Hilden, Germany). To assess the methylation status of nephrin gene promoter, the unmethylated-specific (U) primers including 5′-GGTTGGAAGATTTTATGTTTTTTGA and 5′-CAAACAACTACCCTAAACATCCATA and the methylated-specific (M) primers including 5′-GTTGGAAGATTTTATGTTTTCGA and 5′-GAACAACTACCCTAAACATCCGTA were used in PCRs. The PCR reaction mixture (20 μl) contained 80 ng of bisulfite-treated DNA and 0.5 μM of each primer in 1X EpiTect Master mix. PCR conditions were as follows: 95°C for 10 min; then 35 cycles of 94°C for 15 s, 50°C for 30 s, and 72°C for 30 s; finally 72°C for 10 min.

### Knockdown and overexpression of KDM6A or KLF10 in cultured podocytes

To knock down KDM6A expression in cultured podocytes, cells were transfected with 50 nM of double-stranded KDM6A siRNAs (s75838; Ambion) with sense sequence 5′-GGACUUGCAGCAC GAAUUATT. Scrambled siRNAs were used as a negative control in the corresponding transfection experiments. To overexpress KDM6A in podocytes, the KDM6A expression plasmid (EX-Mm05980-Lv183; GeneCopodia, Rockville, MD) was used in transfection experiments. KLF10 siRNA (s75138; Ambion) with sense sequence 5′-CGUCCA GAGUAAGAAGUCA was used for knockdown of KLF10 in

podocytes, whereas the expression plasmid encoding KLF10 (CD513B-KLF10; System Biosciences) was used for KLF10 overexpression in podocytes.

## Protein extraction and Western blot analysis

Protein extracts from cultured cells or kidney tissues were prepared and manipulated according to our previously described protocols (Lin *et al*, 2010a). Kidney tissues (100–150 mg) were ground in liquid nitrogen with pestle and mortar, homogenized by ultrasonication, and extracted using tissue protein extraction reagent (Pierce, Rockford, IL). Protein concentration was measured by the Bradford method (Bio-Rad) with bovine serum albumin as standard. Equal amounts of protein extracts were separated by SDS–polyacrylamide gel electrophoresis (PAGE). After electrophoresis, the proteins in the gel were transferred onto a polyvinylidene difluoride (PVDF) membrane (Bio-Rad) and subjected to immunoblotting with primary antibodies. Antibodies against nephrin (PA5-20330; Thermo Fisher Scientific; 1:2,000 dilution), KDM6A (sc-514859; Santa Cruz; 1:100 dilution), WT-1 (sc7385; Santa Cruz; 1:750 dilution), H3K27me1 (#7693; Cell Signaling; 1:1,000 dilution), H3K27me2 (#9728; Cell Signaling; 1:1,000 dilution), H3K27me3 (#9733; Cell Signaling; 1:1,000 dilution), Pan-methyl-H3K9 (#4473; Cell Signaling; 1:1,000 dilution), actin (#4970s; Cell Signaling; 1:5,000 dilution), podocin (P0372; Sigma; 1:1,000 dilution), Snail (ab180714; Abcam; 1:1,000 dilution), KLF10 (ab73537; Abcam; 1:1,000 dilution), and synaptopodin (sc51584100; Santa Cruz; 1:1,000 dilution) were obtained commercially. Subsequently, these membranes were probed with a goat anti-rabbit IgG-HRP antibody (sc-2004; Santa Cruz; 1:1,000 dilution) or goat anti-mouse IgG-HRP antibody (sc-2005; Santa Cruz; 1:1,000 dilution), and then immersed with chemiluminescent detection reagents (Pierce) followed by visualized on Hyperfilm ECL film.

## RNA extraction and quantitative reverse transcription (RT)-PCR

Extraction of total RNAs from cultured cells or tissues using QIAzol reagent (Qiagen Inc., Valencia, CA) or Tri reagent (Sigma) according to the manufacturer's instructions. One microgram of total RNA was reverse-transcribed with ReverAid™ M-MuLV reverse transcriptase (Fermentas) to allow synthesis of first-strand cDNA. PCR mixtures (25 μl) containing cDNA template equivalent to 20 ng total RNA, 2.5 μM of specific primer pairs, and 2X iQ™ SYBR Green Supermix (Bio-Rad Laboratories, Hercules, CA) were prepared and subject to PCR amplification using the iCycler iQ® Real-time PCR Detection System (Bio-Rad). The specific primer pairs used for amplification of KDM6A, nephrin, WT-1, KLF10, and actin cDNAs were designed using MIT Primer3 software (primer3.ut.ee). Primer sequences and conditions of PCR amplification were available on request. Relative changes in gene expression detected in quantitative RT–PCR were calculated as described previously (Lin *et al*, 2010a).

## Diabetic animal models and GSK-J4 treatment

Three-month-old male C57BL/6 mice (BioLasco Biotechnology Co., Taiwan) were intraperitoneally given 190 mg/kg streptozotocin (STZ) to induce diabetes. Each diabetic mouse was given 1–2 unit/kg insulin to equalize blood glucose levels as described previously (Lin *et al*, 2010b). Mice with postfasting blood glucose (200–300 mg/dl) were considered as diabetes. For *in vivo* administration of GSK-J4 (4594; Tocris Bioscience, UK), mice were treated daily with 0.4 mg/kg by subcutaneous injection at 1 week after diabetes induction. Normal or diabetic mice were sacrificed by an overdose of sodium pentobarbital at 4, 8, and 12 weeks ($n = 8$ for each) after diabetes. All animal experimental protocols were approved by the Institutional Animal Care and Use Committee of the Chang Gung Memorial Hospital (No. 2015061902) and were performed according to the Animal Protection Law by the Council of Agriculture, Executive Yuan (R.O.C) and the guideline of Nation Institutes of Health (Bethesda, MD). Animal experiments were performed at the Laboratory Animal Center, Department of Medical Research, Chang Gung Memorial Hospital at Chiayi. The Laboratory Animal Center is accredited by the Association for the Assessment and Accreditation of Laboratory Animal Care International (AAALAC) and has a full-time veterinarian. Animals were housed in a room with constant temperature (20–25°C) and humidity (40–60%) under a 12-h light/dark cycle. Mice cages were limited to 2 mice per cage, and animals were given free access to food and water.

## Podocyte-specific *KDM6A* knockout mice and *KLF10* knockout mice

To generate knockout of *KDM6A* specific in podocytes of mice, transgenic podocin-Cre mice [*129S6.Cg-Tg(NPHS2-cre)259Lbh/BroJ*; The Jackson Laboratory] were crossed with *KDM6*^flox mice (B6; *129S-Kdm6 a^{tm1.1Kaig}*; The Jackson Laboratory) to generate podocin-Cre/*KDM6A*^flox mice, designated as *KDM6A*-KO mice in the study. The genotype of the generated *KDM6A*-KO mice was verified by PCR analysis, and the expression of KDM6A in kidney glomeruli or in podocytes was confirmed by using quantitative RT–PCR or Western blot analysis. All *KDM6A*-KO mice were viable and fertile under normal conditions. *KLF10*-KO mice (Yang *et al*, 2013) were kindly provided by Dr. Vincent H.S. Chang (Taipei Medical University, Taiwan), and the genotype and expression of *KLF10* in knockout mice were confirmed in our laboratory. All homozygous *KLF10*-KO mice used in the experiments had a mixed genetic background between B6129 and C57BL/6 over 7 generations. For animal studies, wild-type or knockout mice were randomly allocated to experimental groups.

## Isolation of glomeruli and podocytes

Mouse glomeruli were isolated using the magnetic bead method described by Takemoto *et al* (2002). Briefly, mice were perfused with Dynabeads (M-450)-containing saline through the heart, which resulted in accumulation of Dynabeads in glomerular vessels. Kidneys were then minced into small pieces, digested by collagenase, and filtered through a 100-mesh sterile stainless sieve. Glomeruli trapped by Dynabeads were isolated and gathered by a magnetic particle concentrator. In certain experiments, primary podocytes were further isolated according to the previously reported protocol (Golos *et al*, 2002). Briefly, isolated glomeruli were incubated in flask containing RPMI 1640 for 7 days. Outgrowing epithelial cells, which correspond to podocytes, were trypsinized, filtered through a 33-μm sieve, and then cultured in RPMI1640 with

10% FBS at 37°C. Substantial expression of podocyte-specific markers such as Wilms' tumor-1 protein (WT-1) and nephrin was confirmed in the isolated primary podocytes.

## Urine and blood biochemistry

Urine excretion of each animal was collected using a metabolic cage system. Levels of total protein in urine were measured using the respective kits as described previously (Lin *et al*, 2014). Serum hemoglobin A1c (HbA1c) and blood glucose level in peripheral blood were measured according to the manufacturer's instructions (Primus Diagnostics, Kansas, MO).

## Immunofluorescence

Immunofluorescence staining of the 5-micron-thick kidney sections or cultured podocytes grown on coverslips was performed as described previously (Lin *et al*, 2014). The cultured podocytes on coverslips were fixed in ice-cold trichloroacetic acid for 15 min, or PBS with 4% paraformaldehyde and 4% sucrose for 10 min. Cells were permeabilized in 0.1% Triton X-100 for 10 min, blocked in 10% BSA in PBS for 1 h, and incubated with specific primary antibodies, and then with appropriate fluorophore-conjugated secondary antibodies. Primary antibodies against nephrin (AF3159; R&D Systems; 1:250 dilution), KDM6A (ABE409F; Millipore; 1:500 dilution), WT-1 (sc7385; Santa Cruz; 1:250 dilution), F-actin (A12379; Invitrogen; 1:1,000 dilution), H3K27me3 (#9733; Cell Signaling; 1:500 dilution), KLF10 (ab73537; Abcam; 1:250 dilution), and human KLF10 (MA5-20125; Thermo Fisher Scientific; 1:250 dilution) were purchased commercially. Secondary antibodies, including donkey anti-goat IgG H&L (DyLight® 488) (ab96931; Abcam), donkey anti-mouse IgG H&L (Alexa Fluor® 488) (A21202; Invitrogen), goat anti-rabbit IgG H&L (DyLight® 488) (ab96883; Abcam), goat anti-rabbit IgG H&L (Alexa Fluor® 594) (ab150080; Abcam), and goat anti-mouse IgG H&L (Alexa Fluor® 594) (ab150116; Abcam), were used at 1:250 dilution. Cells were incubated briefly in 4'- 6-diamidino-2-phenylindole (DAPI, Molecular Probes) for nuclei visualization. To quantify the fluorescence intensities of labeled proteins in glomeruli, glomerular areas in kidney sections were routinely discriminated by the unique globular morphology of glomeruli viewed in bright field together with the DAPI-positive staining observed under the same fluorescent microscope settings. Glomerular areas in kidney sections were subsequently marked, and the mean fluorescence intensity per cell within the areas was quantified using the CellSens software package (Olympus).

## RNA sequencing (RNA-Seq) analyses

Total RNAs were prepared from primary podocytes that were infected with control lentiviruses (LPP-NEG-Lv183-050; GeneCopoeia, Rockville, MD) or KDM6A-encoding lentiviruses (LPP-Mm05980-Lv183-050; GeneCopoeia, Rockville, MD) for 48 h. The isolated RNAs were then used for whole-genome RNA next-generation sequencing (RNA-Seq) and analysis performed by Welgene Biotech Co., Ltd (Taipei, Taiwan). Briefly, RNAs were first extracted using Trizol reagent (Invitrogen, USA) according to the manufacturer's instruction. The purified RNAs were quantified at OD260 nm using a ND-1000 spectrophotometer (Nanodrop Technology, USA) and qualified using a Bioanalyzer 2100 (Agilent Technology, USA) with RNA 6000 LabChip kit (Agilent Technology, USA). All procedures for library preparation and sequencing were performed according to the Illumina protocol. Library construction was carried out using the TruSeq RNA Library Preparation Kit for 75-bp single-end sequencing on Solexa platform. The sequence was directly determined by sequencing-by-synthesis technology via the TruSeq SBS kit (Illumina Inc., USA). Raw sequences were obtained from the Illumina Pipeline software bcl2fastq v2.0, which was expected to generate 30 million reads per sample. Qualified reads after filtering low-quality data were analyzed using TopHat/Cufflink. Quantification for gene expression was calculated as fragments per kilobase of transcript per million mapped reads (FPKM). For differential expression analysis, CummeRbund (Illumina Inc., USA) was used to perform statistical analysis of the gene expression profiles. The reference genome and gene annotations were retrieved from the Ensembl database.

## Chromatin immunoprecipitation (ChIP) assay

ChIP assay was performed using the EZ-Magna ChIP A/G kit (17-10086; Millipore) according to the manufacturer's instruction. Briefly, sonicated chromatin complexes were immunoprecipitated using antibodies to KLF10 (ab73537; Abcam; 10 µg per reaction), acetyl-histone H4 (06-866; Millipore; 10 µg per reaction), Dnmt1 (ab13537; Abcam; 10 µg per reaction), and Dnmt3 (ab2850; Abcam; 10 µg per reaction). DNA fragments in the immunoprecipitates were extracted and subjected to quantitative PCR analysis. The PCR primers used in the study were as follows: 5′-CTGGCAGGCAGG GAGGGAGG and 5′-TGCAGCCTGCAAGGCTTCTC for the nephrin promoter region (−2,052/−1,753); 5′-AGGGGGATAGTTCAGACTTC and 5′-GAGGCCTCCTAGGACTCTCT for the nephrin promoter region (−1,802/−1,553).

## Human kidney tissue samples

Patients with unilateral renal masses suspicious for renal cell carcinoma (six subjects with diabetic nephropathy and the other six subjects without diabetes) were retrospectively enrolled in this study. All patients underwent radical nephrectomy and gave their informed consent to participate in the study. The unaffected normal surrounding renal tissues from these patients were obtained from Tissue Bank, Department of Medical Research, Chang Gung Memorial Hospital at Chiayi. All tissue samples were stored at −80°C until use. This study was approved by our local ethic committee (IRB2015061902). Informed consent was obtained from all individual participants included in the study, and the experiments performed with the samples conformed to the principles set out in the WMA Declaration of Helsinki and the Department of Health and Human Services Belmont Report.

## Human urinary exosomal RNA isolation and analysis

Urine exosome RNA was isolated from 5 ml of human cell-free urine using SeraMir Exosome RNA Purification Kit for Urine (RA806TC-1; System Biosciences). Briefly, 1 ml of ExoQuick-TC solution was added to urine samples and mixed gently. After incubation for at 4°

**The paper explained**

**Problem**

Podocyte dysfunction is causally related to the progression of diabetic nephropathy. Despite extensive efforts, there are not enough therapeutic interventions in preventing the progression of diabetic nephropathy, suggesting that additional detrimental pathways for diabetic nephropathy need to be further elucidated. Although a large body of experimental evidence has revealed that epigenetic reprogramming could be a major contributor to podocyte dysfunction in diabetic nephropathy, the detailed linkage between the global epigenetic reprogramming and the resultant signal transduction in podocytes under diabetic conditions remains obscure.

**Results**

The major findings from the present study include the following: (i) KDM6A, a histone lysine demethylase, is up-regulated in high glucose-treated podocytes and in podocytes of diabetic mice, causing podocyte dysfunction *in vitro* and *in vivo*; (ii) podocyte-specific knockout of *KDM6A* in mice significantly suppresses diabetes-induced proteinuria and kidney injury; (iii) KDM6A aggravates diabetic podocyte dysfunction by creating a positive feedback loop through up-regulation of a transcriptional factor KLF10; (iv) KLF10 acts as a key transcription repressor to inhibit multiple podocyte-specific markers in podocytes; (v) KLF10 can conversely activate KDM6A expression; (vi) KLF10 inhibits nephrin expression by directly binding to the gene promoter along with the recruitment of methyltransferase Dnmt1; (vii) global knockout of *KLF10* in mice protects against diabetic kidney injury; (viii) as compared to the control subjects, human diabetic nephropathy patients exhibit higher levels of *KDM6A* and *KLF10* proteins or mRNAs in kidney tissues or in urinary exosomes.

**Impact**

Our findings highlight the importance of a reinforcing feedback mechanism involving KDM6A and KLF10 in promoting podocyte dysfunction under diabetic conditions and strongly suggest that targeting the KDM6A–KLF10 feedback loop may be beneficial to protect against diabetes-induced proteinuria and kidney injury.

for 6 h, the mixture was centrifuged at 12,000 *g* for 2 min, and the supernatant was removed by aspiration. Exosomes in the pellet were lysed, and then, RNA was extracted according to the manufacturer's instruction. The individual primer sequences used in quantitative RT–PCR included the following: 5′-ATGGAA ACGTACCTT ACCTG and 5′-ATTAGGACCTGCCGAATGTG for KDM6A; 5′-TCAC ATCTGTAGCCACCCAGGATG and 5′-CTTTCCAGCTACAGCTGAAA GGCT for KLF10; 5′-TCACTACCCCAGGTCTCCAC and 5′-CCCTGCC TCTGTCTTCTCTG for nephrin; 5′-GTAACCCGTTGAACCCCATT and 5′-CCATCCAATCGGTAGTAGCG for 18S rRNA. Relative mRNA levels in urinary exosomes were normalized to 18S rRNA.

**Statistical analyses**

All values were expressed as mean ± SEM. *In vitro* experimental data were collected from at least three independent experiments. Wilcoxon two-sample test was used to evaluate differences between the sample of interest and its respective control. Parametric ANOVA and a Bonferroni *post hoc* test were used to analyze the differences among various treated groups. Data collection and statistical analyses were performed using SPSS version 18.0. *P* < 0.05 was considered statistically significant.

# Data availability

RNA-seq data from this study have been deposited in the Array-Express database and assigned the identifier accession number E-MTAB-7695 (http://www.ebi.ac.uk/arrayexpress/experiments/E-MTAB-7695).

**Expanded View** for this article is available online.

## Acknowledgements

We greatly appreciate Vincent H.S Chang (Taipei Medical University) for providing KLF10 knockout mice. We acknowledge Sun-Sen Yang (National Defense Medical Center) for discussing the generation of knockout mice and for technical assistance. We thank Laboratory Animal Center, Department of Medical Research, Chang Gung Memorial Hospital at Chiayi, and National Laboratory Animal Center for animal breeding and sample preparation. We acknowledge Tissue Bank, Department of Medical Research, Chang Gung Memorial Hospital at Chiayi for providing human tissue specimens and urinary samples from subjects with diabetes or non-diabetes. The work was supported by MOST grants 104-2314-B-182A-068-MY3 and 107-2314-B-182A-029-MY3 from the Ministry of Science and Technology of Taiwan and by medical research grants CMRPG6E0381~3 and CMRPG6H0361 from Chang-Gung Memorial Hospital at Chiayi, Taiwan.

## Author contributions

C-LL, Y-CH, and P-JC designed and performed experiments, and wrote the paper; Y-TH and Y-HS contributed to data analysis; C-LL, Y-TH, and Y-HS participated in formulating aspects of study design relating to knockout mice; W-CC and C-JW contributed to the interpretation of data and manuscript revision.

## Conflict of interest

The authors declare that they have no conflict of interest.

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
