## [Review Process File · EMBO Molecular Medicine]

A KDM6A–KLF10 reinforcing feedback mechanism aggravates diabetic podocyte dysfunction

Chun-Liang Lin, Yung-Chien Hsu, Yu-Ting Huang, Ya-Hsueh Shih, Ching-Jen Wang, Wen-Chih Chiang, Pey-Jium Chang

Review timeline:	Submission date:	19 September 2018
	Editorial Decision:	24 October 2018
	Revision received:	6 January 2019
	Editorial Decision:	30 January 2019
	Revision received:	9 March 2019
	Accepted:	14 March 2019

Editor: Lise Roth

Transaction Report:

1st Editorial Decision

24 October 2018

Thank you for the submission of your manuscript to EMBO Molecular Medicine. We have now heard back from the three referees whom we asked to evaluate your manuscript. As you will see from the reports below, while they all mention the interest of the study, they also raise substantial concerns on your work, which should be convincingly addressed in a major revision of the present manuscript. In particular, there is a need to further strengthen the data to fully support the conclusions, and to increase the level of mechanistic understanding (crosstalk KLF10/KDM6, role of TGFβ1). I wish to add however, that given the considerable amount of time that would be needed to repeat the *in vivo* experiment in a type 2 diabetic (T2D) model (as requested by referees #1 and #2), these experiments will not be required further consideration of your manuscript (unless you have results readily available).

Addressing the reviewers' concerns in full (with the exception of T2D mouse model experiments) will be necessary for further considering the manuscript in our journal. EMBO Molecular Medicine encourages a single round of revision only and therefore, acceptance or rejection of the manuscript will depend on the completeness of your responses included in the next, final version of the manuscript. For this reason, and to save you from any frustrations in the end, I would strongly advise against returning an incomplete revision and would also understand your decision if you choose to rather seek rapid publication elsewhere at this stage. Should you find that the requested revisions are not feasible within the constraints outlined here and prefer, therefore, to submit your paper elsewhere, we would welcome a message to this effect.

EMBO Molecular Medicine has a "scooping protection" policy, whereby similar findings that are published by others during review or revision are not a criterion for rejection. Should you decide to submit a revised version, I do ask that you get in touch after three months if you have not completed it, to update us on the status. Please also contact us as soon as possible if similar work is published elsewhere. If other work is published, we may not be able to extend the revision period beyond three months.

I look forward to receiving your revised manuscript.

***** Reviewer's comments *****

Referee #1 (Remarks for Author):

Lin et al reported the potential role of KLF10-KDM6A axis in the development of podocyte injury in diabetic condition. Basically, KDM6A suppression restored the levels of HG-suppressed podocyte markers such as Nephron and WT1. Also Nephron promoter methylation was induced by HG but reduced by KDM6A inhibition. KLF10 is key to understand KDM6A induction. Also authors should that these findings were relevant in vivo diabetic mice and also human samples. Study was done with proper design and also logically correct. This reviewer would like to confirm followings.

- 1) The most important point that this reviewer needed to confirm was the inter-relation between KLF10 and KDM6A. Authors found that possible bidirectional positive feedback between these two molecules. However pathological aspect, which molecule could play dominant role to induce these detrimental effects on podocyte biology was not clear yet. For that experiment, authors must perform additional analysis that 1) KDM6A knockdown cells with KLF10 overexpression, and KLF10 knockdown with KDM6A overexpression. In these cells condition authors should analyze all podocyte biology.
- 2) Is similar phenomena could expected in type 2 diabetic mice models?
- 3) In vivo analysis, authors should show urine albumin and also renal functional data such as cystatin C etc.
- 4) Also blood pressure needed.
- 5) Also Electron microscope analysis should be performed.
- 6) Fig 1A, all gene expression analyze were shown, but authors should analyze other KDM protein levels by western blot.
- 7) Was TGF- β 1 important for KDM6A-induced by KLF10?
- 8) Also what is the role of TGF- β 1, in KLF-10-induced KDM6A induction? Authors can perform neutralizing experiments for 7),8). Also smad inhibitor to confirm cell target cell signal events.
- 9) Can TGF- β explain high glucose induced KLF10 and KDM6A? Also neutralizing and smad inhibitor experiment can be done.

Referee #2 (Remarks for Author):

In the present study, Lin CL and colleagues describe a new pathway driven by KDM6A and KLF10 in the progression of podocyte injury in diabetic nephropathy. Unfortunately, the data provided here do not convincingly support this major conclusion and better data are needed.

Major comments:

1. The experimental model used in the study was streptozotocin-induced mouse model. However, podocytes express insulin receptors. What is the status of KDM6A and KLF10 expression in type 2 diabetes models in which insulin is present?
2. Please show KDM6A and KLF10 expression in diabetes using immunohistochemistry. Please show it expression in podocytes with colocalization with podocyte using a marker such as WT1, Synaptopodin...
3. Targeting KDM6A in podocytes could attenuate diabetes-induced kidney injury. To demonstrate that the authors should check mechanisms such as glomerular and interstitial fibrosis. Same for KLF10-KO mice.
4. Podocyte apoptosis is a feature of diabetic nephropathy. Is there a podocyte number reduction in wild-type mice compared with KDM6A-KO and KLF10-KO mice with diabetes? In these experiments and at the time point studied, authors should measure glomerular TUNEL staining to detect or not evidence of apoptosis specifically within glomerular podocytes. Also immunofluorescence microscopy for caspase-3 cleavage is needed. How does GSK-J4 treatment affect to apoptosis?

5. Figure 4.C. No significant differences in body weights or blood glucose levels between the wild-type and KDM6A-KO mice with diabetes were observed during the 8-week experimental period. Why in the KDM6A-KO mice the DM treatment last no more than 8 weeks unlike in normal mice in Figure 3, which lasts 12 weeks? Please explain that difference of time in the treatment.

6. Figure 4.D. Please explain the non-differences in HbA1c between wild-type mice with diabetes and KDM-KO mice with diabetes? Please comment in the text.

7. When does increased KDM6A expression start in the STZ injected mice? Immunohistochemistry showing KDM6A increasing is needed. How does it correlate with the development of proteinuria? How does KDM6A increasing correlate with Nephlin decreasing?

8. Regarding the human studies, the author should include a table with some data: type 1 or 2 diabetes, use of RAS blockers, use of SGLT2 inhibitors, eGFR, albuminuria, HbA1C...

Minor changes:

1. Figure 1.B legend. Please correct "popdocytes"
2. Figure 4.I. Please, explain why the authors look at Snail in the western blot. It is not mentioned in the results.

Referee #3 (Remarks for Author):

Lin et al. identified a positive feedback loop between KDM6A-KLF10 in podocytes exposed to hyperglycemic condition, which suppresses a number of podocyte-specific genes. Their work highlights a potential new therapeutic target to preserve podocyte function in diabetes mellitus. The following questions need to be address to improve the manuscript.

- 1) the majority of the western blot results lack corresponding statistical analyses. These must be analyses and included in the manuscript.
- 2) the immunofluorescence staining also require quantitative analyses and statistics.
- 3) The authors claim that H3K27me2 was decreased in HG-treated podocytes, but the data in Fig 1G do not seem to support this conclusion.

1st Revision - authors' response

6 January 2019

Referee #1 (Remarks for Author):

Reviewer's Summary: "Lin et al reported the potential role of KLF10-KDM6A axis in the development of podocyte injury in diabetic condition. Basically, KDM6A suppression restored the levels of HG-suppressed podocyte markers such as Nephlin and WT1. Also Nephlin promoter methylation was induced by HG but reduced by KDM6A inhibition. KLF10 is key to understand KDM6A induction. Also authors showed that these findings were relevant in vivo diabetic mice and also human samples. Study was done with proper design and also logically correct. This reviewer would like to confirm followings."

Response: We appreciate the reviewer's positive evaluation of our work. Detailed responses to the reviewer's concerns and suggestions are included in the following.

Comment 1: "The most important point that this reviewer needed to confirm was the inter-relation between KLF10 and KDM6A. Authors found that possible bidirectional positive feedback between these two molecules. However pathological aspect, which molecule could play dominant role to induce these detrimental effects on podocyte biology was not clear yet. For that experiment, authors must perform additional analysis that 1) KDM6A knockdown cells with KLF10 overexpression, and KLF10 knockdown with KDM6A overexpression. In these cells condition authors should analyze all podocyte biology."

Response: We thank the reviewer for this suggestion. As recommended by the reviewer, we have done the experiments regarding KDM6A overexpression in combination with KLF10 knockdown,

and KLF10 overexpression in combination with KDM6A knockdown in cultured podocytes. Results from these experiment are now provided in **Expanded View Figure EV4**, showing that down-regulation of podocyte-specific marker proteins (such as nephrin and WT-1) is strongly correlated with increased KLF10, but not increased KDM6A. This important point is also added to the Result section on **page 13** in the revised manuscript.

Comment 2: “Is similar phenomena could expected in type 2 diabetic mice models?”

Response: Although further understanding of the KDM6A-KLF10 detrimental signaling in a type 2 diabetic mice model is important, we think that this would be a future direction. Despite the lack of direct evidence for the involvement of an active KDM6A-KLF10 signaling axis in type 2 diabetes, analysis of human samples in the present study has revealed that diabetic patients (**all with type 2 diabetes**) did have higher mRNA levels of KDM6A and KLF10 in urinary exosomes as compared to control subjects (Figure 7). The baseline characteristics of human control subjects and diabetic patients enrolled in the present study are now provided in **Appendix Table S1**.

Comment 3: “In vivo analysis, authors should show urine albumin and also renal functional data such as cystatin C etc.”

Response: To further strengthen our main conclusions, we have now included several additional *in vivo* data and analyses in the revised manuscript (**Extended View Figures EV1 and EV3 & Appendix Figure S2**). Data on the urinary levels of albumin and cystatin C are now provided in **Extended View Figures EV1A, EV1C, EV3A and EV3C** as well as **Appendix Figure S2A**.

Comment 4: “Also blood pressure needed.”

Response: Data on systolic blood pressure of experimental mice are now included in **Extended View Figures EV1B and EV3B**.

Comment 5: “Also Electron microscope analysis should be performed.”

Response: Data on electron microscope analysis of glomerular basement membrane (GBM) thickening in renal tissues of experimental mice (normal, diabetic, KDM6A-KO or KLF10-KO mice) are now provided in **Extended View Figures EV1E and EV3E**.

Comment 6: “Fig 1A, all gene expression analyze were shown, but authors should analyze other KDM protein levels by western blot.”

Response: In Fig. 1E (not in Fig 1A), we screened the potential KDM genes involved in high glucose-mediated podocyte dysfunction by quantitative RT-PCR. Although we did not examine all KDM protein levels in Western blotting in the case, we did confirm the increased protein expression of the selected KDM gene (KDM6A) in the following experiments.

Comment 7: “Was TGF- β 1 important for KDM6A-induced by KLF10?”

Response: We thank the reviewer for asking the question regarding the potential role of TGF- β 1 in the positive inter-regulation between KDM6A and KLF10. We have done the experiments using KDM6A or KLF10 overexpression along with addition of TGF- β 1-neutralizing antibody in cultured podocytes. In the experiments, we show that neutralization of TGF- β 1 with specific antibody could not influence the positive inter-regulation of these two proteins. These results are now found in **Extended View Figure EV5B and 5C** in the revised manuscript. Additionally, we have added an explaining paragraph in the text on **pages 13-14** to response the comment.

Comment 8: “Also what is the role of TGF- β 1, in KLF-10-induced KDM6A induction? Authors can perform neutralizing experiments for 7),8). Also smad inhibitor to confirm cell target cell signal events.”

Response: In the revised manuscript, additional experiments using KDM6A or KLF10 overexpression in combination with addition of TGF- β 1 in cultured podocytes are provided (**Extended View Figure EV5B and 5C**). Our results show that TGF- β 1 is not essential for the

positive inter-regulation between KDM6A and KLF10. In the revised manuscript, an explaining paragraph is included on **pages 13-14**.

Comment 9: “Can TGF- β explain high glucose induced KLF10 and KDM6A? Also neutralizing and smad inhibitor experiment can be done.

Response: We have now provided additional experiments (**Extended View Figure EV5A**) to show that neutralization of TGF- β 1 by a specific antibody (10 μ g/ml) sufficiently blocks high glucose-mediated upregulation of KDM6A and KLF10. However, the same amount of TGF- β 1-neutralizing antibody could not influence the positive inter-regulation between KDM6A and KLF10 (**Extended View Figures EV5B and 5C**). A model for the relationship between TGF- β 1 signaling and the KDM6A-KLF10 feedback loop in podocytes is now proposed in **Extended View Figure EV5D**. In the revised manuscript, an explaining paragraph is included on **pages 13-14**.

Referee #2 (Remarks for Author):

Reviewer’s Summary: “In the present study, Lin CL and colleagues describe a new pathway driven by KDM6A and KLF10 in the progression of podocyte injury in diabetic nephropathy. Unfortunately, the data provided here do not convincingly support this major conclusion and better data are needed.”

Response: Detailed responses to the reviewer’s concerns are included in the following.

Major Comment 1: “The experimental model used in the study was streptozotocin-induced mouse model. However, podocytes express insulin receptors. What is the status of KDM6A and KLF10 expression in type 2 diabetes models in which insulin is present?”

Response: Despite the lack of direct evidence for the involvement of an active KDM6A-KLF10 signaling axis in type 2 diabetes, analysis of human samples in the present study has revealed that patients with type 2 diabetes did have higher levels of KDM6A and KLF10 mRNAs in urinary exosomes as compared to control subjects (Figure 7). In the revised manuscript, the baseline characteristics of human control subjects and diabetic patients enrolled in the present study are now provided in **Appendix Table S1**. Although it is important to further understand the status of KDM6A and KLF10 in type 2 diabetes models in which insulin is present, we think that this would be a future direction.

Major Comment 2: “Please show KDM6A and KLF10 expression in diabetes using immunohistochemistry. Please show its expression in podocytes with colocalization with podocyte using a marker such as WT1, Synaptopodin...”

Response: In Fig. 6F (confocal immunofluorescence), we have already shown that both increased KLF10 and KDM6A are co-localized in glomerular cells of diabetic wild-type mice. Based on our current hypothesis, increased expression of KLF10 and KDM6A would be correlated with poor expression of podocyte-specific markers such as nephrin, WT1 and synaptopodin in renal glomerular cells. Therefore, it would be not common to see intensive signals of KLF10 and KDM6A co-localizing with podocyte-specific markers in diabetic podocytes.

Major Comment 3: “Targeting KDM6A in podocytes could attenuate diabetes-induced kidney injury. To demonstrate that the authors should check mechanisms such as glomerular and interstitial fibrosis. Same for KLF10-KO mice.”

Response: To further support our main conclusions, we have now provided several additional *in vivo* data and analyses in the revised manuscript (**Extended View Figures EV1 and EV3**). Data on periodic acid-Schiff (PAS) staining of renal tissue sections from experimental mice are included in **Extended View Figures EV1F** (KDM6A-KO mice) and **EV3F** (KLF10-KO mice). The text has also been modified on **pages 9, 12 and 13** to reflect the comment.

Major Comment 4: “Podocyte apoptosis is a feature of diabetic nephropathy. Is there a podocyte number reduction in wild-type mice compared with KDM6A-KO and KLF10-KO mice with

diabetes?

In these experiments and at the time point studied, authors should measure glomerular TUNEL staining to detect or not evidence of apoptosis specifically within glomerular podocytes. Also immunofluorescence microscopy for caspase-3 cleavage is needed. How does GSK-J4 treatment affect to apoptosis?"

Response: As requested by reviewers, we have now provided several additional *in vivo* assays including glomerular TUNEL analysis in the revised manuscript. Data on TUNEL analysis of renal tissues of experimental mice are included in **Extended View Figures EV1D** (KDM6A-KO mice) and **EV3D** (KLF10-KO mice) as well as **Appendix Figure S2B** (GSK-J4-treated diabetic mice). The text has also been modified on **pages 8, 9, 12 and 13** to reflect the comment.

Major Comment 5: "Figure 4.C. No significant differences in body weights or blood glucose levels between the wild-type and KDM6A-KO mice with diabetes were observed during the 8-week experimental period. Why in the KDM6A-KO mice the DM treatment last no more than 8 weeks unlike in normal mice in Figure 3, which lasts 12 weeks? Please explain that difference of time in the treatment."

Response: The 8-week experimental period is commonly used for evaluating therapeutic effects on mice with STZ-induced diabetes. To test for the consistency of our animal model and evaluate appropriate time points for studies, we have attempted to extend the experimental period from 8 weeks to 12 weeks in our initial studies with GSK-J4 treatment (Figure 3). Based on our experiences, we think that the 8-week experimental period would be suitable for our following studies (**non-therapeutic research studies**).

Major Comment 6: "Figure 4.D. Please explain the non-differences in HbA1c between wild-type mice with diabetes and KDM-KO mice with diabetes? Please comment in the text."

Response: HbA1c, also known as glycated hemoglobin, is a marker commonly used to measure long-term blood glucose levels. In the present study, we have found that there is no difference in blood glucose levels between diabetic wild-type mice and diabetic KDM6A-KO mice (Fig 4C). Consistently, no significant difference in HbA1c levels between diabetic wild-type mice and diabetic KDM6A-KO mice was observed (Fig. 4D). This point is now added to the Result section on **page 9** in the revised manuscript.

Major Comment 7: "When does increased KDM6A expression start in the STZ injected mice? Immunohistochemistry showing KDM6A increasing is needed. How does it correlate with the development of proteinuria? How does KDM6A increasing correlate with Nephlin decreasing?"

Response: According to our results shown in Figure 3A (quantitative RT-PCR) and 3B (Western blot analysis), increased KDM6A expression along with decreased nephlin expression in kidney glomeruli might occur within 4 weeks after diabetic induction with STZ.

Major Comment 8: "Regarding the human studies, the author should include a table with some data: type 1 or 2 diabetes, use of RAS blockers, use of SGLT2 inhibitors, eGFR, albuminuria, HbA1C..."

Response: The baseline characteristics of control subjects and patients with diabetic nephropathy enrolled in the study are now provided in **Appendix Table S1**.

Minor Comment 1: "Figure 1.B legend. Please correct "podocytes" "

Response: We thank the reviewer for pointing out this error. It has been corrected.

Minor Comment 2: "Figure 4.I. Please, explain why the authors look at Snail in the western blot. It is not mentioned in the results."

Response: Since the transcription factor Snail was previously implicated in the repression of nephlin expression in glomerular epithelial cells (Matsui et al., 2007), the association between

KDM6A and Snail in regulating nephrin expression was also examined in the study. However, we could not find a correlation between Snail expression and KDM6A-mediated nephrin downregulation in the experiments. This point is now added to the Result section on **page 10** in the revised manuscript.

Referee #3 (Remarks for Author):

Reviewer's Summary: "Lin et al. identified a positive feedback loop between KDM6A-KLF10 in podocytes exposed to hyperglycemic condition, which suppresses a number of podocyte-specific genes. Their work highlights a potential new therapeutic target to preserve podocyte function in diabetes mellitus. The following questions need to be address to improve the manuscript."

Response: We thank the reviewer for the positive comment. Detailed responses to the reviewer's suggestions are included in the following.

Comment 1: "the majority of the western blot results lack corresponding statistical analyses. These must be analyses and included in the manuscript."

Response: Quantitative analyzes and statistics of Western blots are now included in revised **Figures 1-6**.

Comment 2: "the immunofluorescence staining also require quantitative analyses and statistics."

Response: Quantitative analyses and statistics of immunofluorescence staining in the studies are now provided in revised **Figures 3-6**.

Comment 3: "The authors claim that H3K27me2 was decreased in HG-treated podocytes, but the data in Fig 1G do not seem to support this conclusion."

Response: We have redone the experiment regarding the effect of high glucose on levels of H3K27me2 in podocytes, and the results are found in revised **Figure 1G**. New data consistently show that HG-treated podocytes display reduced levels of H3K27me2 as compared to normal controls.

2nd Editorial Decision

30 January 2019

Thank you for the submission of your revised manuscript to EMBO Molecular Medicine. We have now received the enclosed reports from the three referees that were asked to re-assess it. While they are mostly satisfied with the revisions of the manuscript, referees 2 and 3 still have a few concerns that should be addressed before acceptance of your manuscript. Please address the following:

1) Referees' comments:

- Please provide immunohistochemistry data as requested by referee 2 and previously mentioned in the first review of the manuscript.
- Please clearly describe the immunofluorescence quantification methods as requested by referee 3.

***** Reviewer's comments *****

Referee #1 (Comments on Novelty/Model System for Author):

Significant informaton based on highly original idea

Referee #1 (Remarks for Author):

Done well. Authors did all the requirements from this reviwer

Referee #2 (Remarks for Author):

In the present study, Lin CL and colleagues describe a new pathway driven by KDM6A and KLF10 in the progression of podocyte injury in diabetic nephropathy. Although the new data provided have improved the quality of the manuscript, there are still some major comments that have not been addressed.

Major Comments:

Immunohistochemistry showing KDM6A increasing in the STZ injected mice is needed.
How does it correlate with the development of proteinuria?
How does KDM6A increasing correlate with Nephlin decreasing?

Referee #3 (Remarks for Author):

The revision has included appropriate quantification for the majority of the analyses, but additional clarification is needed for some experiment and quantitative analyses.

- 1) Fig 4F: how was the H3K27me signal quantified? I don't see a glomerular podocyte marker as a counter staining.
- 2) Fig 5F, same question, no podocyte marker as a counter staining, how were KLF10 and KDM6A quantified? these genes are not only expressed by podocytes, given the extra-glomerular staining.
- 3) Fig 6A: no podocyte markers either. please describe in the methods or legend, how the quantification was carried out.
- 4) Fig. 3D, I believe the GSK-J4 treated group should be compared to the DM mice. so is this * a typo?

2nd Revision - authors' response

9 March 2019

Referee #1 (Comments on Novelty/Model System for Author):

Significant information based on highly original idea

Referee #1 (Remarks for Author):

Done well. Authors did all the requirements from this reviewer.

Response: We sincerely thank the reviewer for providing insightful comments contributing to the improvement of the manuscript.

Referee #2 (Remarks for Author):

In the present study, Lin CL and colleagues describe a new pathway driven by KDM6A and KLF10 in the progression of podocyte injury in diabetic nephropathy. Although the new data provided have improved the quality of the manuscript, there are still some major comments that have not been addressed.

Major Comments:

*Immunohistochemistry showing KDM6A increasing in the STZ injected mice is needed.
How does it correlate with the development of proteinuria?
How does KDM6A increasing correlate with Nephlin decreasing?"*

Response: We have provided additional data (Appendix Fig S2) as requested by the reviewer in the revised manuscript. In the additional Appendix Fig S2, we include i) the results of double immunofluorescence staining (KDM6A & nephrin) of kidney sections that were obtained from normal mice and the 4-, 8- and 12-week diabetic mice, and ii) urinary protein excretion at 4, 8 and 12 weeks in normal and diabetic mice. Our results consistently show that both increased KDM6A expression and reduced nephrin expression were closely associated with the presence of proteinuria in the 4-, 8- and 12-week diabetic mice. We have also added a short paragraph to the "Results" section (page 8) to describe the experiments.

Referee #3 (Remarks for Author):

The revision has included appropriate quantification for the majority of the analyses, but additional clarification is needed for some experiment and quantitative analyses.

Comment 1: *Fig 4F: how was the H3K27me signal quantified? I don't see a glomerular podocyte marker as a counter staining.*

Response: To quantify the fluorescence intensities of labeled proteins in Fig 4F (H3K27me3) [or Fig 6F (KDM6A & KLF10) and Fig 7A (human kidney samples) described in the following comments], glomerular areas in kidney sections have to be first determined. In all our immunofluorescence analysis, glomerular areas in kidney sections were routinely discriminated by the unique globular morphology of glomeruli in bright field (see Fig. A & B; “a” panels) together with the DAPI-positive staining (see Fig. A & B; “b” panels) observed under the same fluorescent microscope settings. Glomerular areas in kidney sections were subsequently marked and the mean fluorescence intensity per cell within the areas were quantified using the CellSens software package (Olympus). This important point is now added to the “Materials & Methods (Immunofluorescence)” section on page 27.

Comment 2: *Fig 5F, same question, no podocyte marker as a counter staining, how were KLF10 and KM6A quantified? these genes are not only expressed by podocytes, given the extra-glomerular staining.*

Response: Despite the lack of a podocyte marker as a counter staining, the expression of KDM6A and KLF10 within glomerular areas (Fig 6F) could be quantified based on the method described above (see Response to Comment 1 and Fig. A & B).

Comment 3: *Fig 6A: no podocyte markers either. please describe in the methods or legend, how the quantification was carried out.*

Response: Although there are no podocyte markers in some experiments of Fig 7A, the fluorescence intensity of each labeled protein within glomeruli could be quantified based on the same method described above (see Response to Comment 1 and Fig. A & B on the next page).

Comment 4: *Fig. 3D, I believe the GSK-J4 treated group should be compared to the DM mice. so is this * a typo?*

Response: In Fig. 3D (HbA1c), we confirm that the information given in the figure is correct. Compared to the normal control group, both the untreated and GSK-J4-treated DM groups significantly exhibited an elevated HbA1c level (* $P=0.0082$ and * $P=0.0096$, respectively). However, we did not find a statistically significant difference in HbA1c levels between the untreated DM group and the GSK-J4-treated group ($P=0.1044$; not labelled with the symbol “#”). The exact P values for Fig. 3D have been provided in Appendix Table S3 (page 12).

Fig. A & B: Examples for locating glomerular areas in kidney sections. (a) bright-field images; (b) images for DAPI staining; (c) images for nephrin staining; (d) merged images.

Corresponding Author Name: Pey-Jium Chang

Journal Submitted to: EMBO Mol Med

Manuscript Number: EMM-2018-09828